# Land use is a stronger determinant of ecological network complexity than the number of trophic levels

**Angela R. Amarillo-Suárez**[1]*, **Mariana Camacho-Erazo**[2], **Henri W. Herrera**[2]

**1** Departamento de Ecología y Territorio, Facultad de Estudios Ambientales y Rurales, Pontificia Universidad Javeriana, Bogotá, Cundinamarca, Colombia, **2** Escuela de Ingeniería en Recursos Renovables, Facultad de Recursos Naturales, Escuela Superior Politécnica del Chimborazo, Riobamba, Chimborazo, Ecuador

* aamarillo@javeriana.edu.co

**Data Availability Statement:** The data are held in the following repository: https://www.ebi.ac.uk/biostudies/studies/S-BSST1313.

## Abstract

Land modification causes biodiversity loss and ecosystem modification. Despite many studies on the impacts of this factor, there is little empirical evidence on how it affects the interaction networks of plants, herbivores and their natural enemies; likewise, there is little evidence on how those networks change due to differences in the complexity of the communities they comprise. We analyzed the effects of land use and number of trophic levels on the interaction networks of exotic legume species and their associated arthropods. We collected seedpods from five exotic legume species (one of them invasive) in four land use types (urbanization, roadside, *L. leucocephala* plantation, wooded pasture) on Santa Cruz Island in the Galapagos, and obtained all arthropods that emerged from the seeds. Then, we built and analyzed the interaction networks for each land use at two community scales, each with different numbers of trophic levels: (1) three levels: plant-seed beetle-parasitoid (PSP), and (2) more than three levels: plant-seed beetle-parasitoid-predator and other trophic guilds (PSPP). Land use was more relevant than number of trophic levels in the configuration of species interactions. The number of species and interactions was highest on roadsides at PSPP and lowest in plantations at PSP. We found a significant effect of land use on connectance and interaction evenness (IE), and no significant effect of number of trophic levels on connectance, diversity or IE. The simultaneous analysis of land use and number of trophic levels enabled the identification of more complex patterns of community structure. Comparison of the patterns we found among islands and between exotic and native legumes is recommended. Understanding the structure of the communities analyzed here, as well as the relative contribution of their determinants of change, would allow us to develop conservation plans according to the dynamics of these neo-ecosystems.

## Introduction

Humans are increasingly modifying ecosystems worldwide to the extent that some scientists believe that such anthropogenic changes have shaped the beginning of a new geological era,

**Funding:** This study was supported by the Research Institute of the Escuela Superior Politécnica de Chimborazo (ESPOCH), Riobamba, in the form of a grant to HWR [301-CP-218] and Pontificia Universidad Javeriana in Bogotá, Colombia in the form of project funding to ARA-S [00008226]. This article is contribution 2565 of the Charles Darwin Foundation for the Galapagos Islands and number 006 under cooperative agreements ESPOCH-FCD/2017-2022 and ESPOCH-GNPD/2018-2021.

**Competing interests:** The authors have declared that no competing interests exist.

the Anthropocene [1, 2]. Setting aside the controversy elicited by this statement, there is a consensual evidence that humans have a role in the diminishing biodiversity at different scales. This includes changes in landscape structure and land use [3, 4], as well as reductions in functional and taxonomic biodiversity [5] and their interactions [6, 7] with the resulting loss and extinction of local and global species [8, 9].

The Millennium Ecosystems Evaluation [10] implicates fragmentation, habitat loss and exotic species introductions as the main causes of declines in biodiversity. In addition to the negative impacts on taxonomic diversity [11, 12] these three factors modify the structure of biological communities [13, 14]. Plant-herbivore interactions involve approximately 40% of biodiversity [15] and are some of the most important mechanisms for the bottom-up flow of matter and energy in ecosystems [16]. In addition, plant-herbivore interactions are responsible for the high diversification of insects and plants [17] and their broad distribution, as well as the structures of local communities [16]. There is heterogeneity in the data sets and methods that have been used used to study plant–herbivore interactions. Understanding how the structure of these relationships functions, changes, and is constituted, as well as how it is affected by anthropogenic activities, is necessary for understanding the cascading impacts on ecosystem integrity.

Ecological network analysis has been recently used to describe and evaluate the diversity of species, ecological processes, and the relationships between community structure and functioning [18–20]. Contrary to traditional descriptions of communities based on alpha and beta diversity, this method of analyzing communities enables a systemic understanding of the structure, organization, and composition of species and the complexity of their interactions. Ecological networks are dynamic and change in response to time [21], seasonality, biological cycles, and anthropogenic perturbations such as changes in land use [22, 23] and introductions of exotic species causing cascading effects on nutrient cycling [20, 24] and the structure of native communities [13, 15, 25]. Such perturbations can even push populations and species to extinction [26, 27]. Ecological network analysis also shows the direct relationship between species numbers and the stability of the community; the more species that are connected in the community, the greater the robustness and connectivity, generating a more stable community [25, 28–30].

Ecological network parameters are negatively affected by anthropogenic factors in a manner similar to how species diversity is negatively affected [30, 31]. However, very few studies have validated this prediction, and some results have been contradictory [15]. For example, a study conducted in forest fragments in Argentina demonstrated that connectance, network size, and species richness diminished as the size of forest fragments decreased, while there was no border effect for such parameters [32, 33]. However, a different study showed that connectance increased in perturbed ecosystems as a result of a decrease in specialist species [19]. Such differences in a few studies highlight the importance of conducting a larger number of studies to determine patterns of change in the networks and in interactions.

The Galapagos Islands are an ideal model for analyzing the effects of anthropogenic perturbations on interaction networks. This highly diverse place preserves a large number of native and endemic species [27]. However, the introduction of exotic species has increased due to human activities such as cattle ranching, agriculture, and increasing tourism [34, 35] which have transformed natural areas [33] and are of great concern to environmental institutions in charge of the preservation and sustainable use of fragile and complex communities in the Galapagos archipelago [36]. The archipelago contains approximately 900 exotic species of plants, which is much higher than the 522 species of native and endemic species [27, 37]. Of the 900 exotic species, 30 are invasive [37] and are a major threat to the species diversity on the islands [38, 39]. Within the arid zone of the Galapagos, the second larger area after volcanic rock,

legumes are the most conspicuous and diverse group of plants [40–42], with 82 species [43, 44]. Of these, 67 are exotic [37, 45]. Legume plants interact with other organisms, such as animals that feed on their leaves and fruits. In the Galapagos, 12 species of Bruchinae insects feed on the seeds of 15 legume species, and many of these insects play an important role in seed dispersion and viability [25, 46] and establish more extensive communities along with their natural enemies and competitors. Seed predator-legume plant interactions provide excellent models for analyzing how anthropogenic changes affect interactions among organisms. Seed predators are easy to identify, easy to maintain under laboratory conditions and have a restricted distribution. In addition, because the development of herbivores and parasitoids occurs inside the seeds, the accuracy of the information gathered is higher than that obtained when examining free-leaving herbivorous organisms.

Studies analyzing how networks parameters and structure change when simultaneously considering the number of trophic levels and a variety of land uses are important for testing the multifactorial causes determining the responses of biodiversity to environmental change. Thus, the objective of this study was to analyze the effect of land use and number of trophic levels on the structure and complexity of the interaction networks between arthropods and plants in the arid zone of Santa Cruz Island in Galapagos, Ecuador. These networks comprised the association of the seeds of five species of exotic leguminous trees with arthropods. To do this, we defined two communities that differ in the number of trophic levels and on four land uses: the first community was composed of legume, seed beetles, and Hymenoptera parasitoids (hereafter called PSP); and the second community was composed of the PSP group plus predators and other phytophagous species (hereafter called PSPP). The four land uses were roadside, urbanization, wooded pasture, and plantation.

## Materials and methods

### Study site

The Galapagos Archipelago has 13 large islands, six of medium size, and many small islets. The archipelago is located 1000 km off the coast of continental Ecuador. Ecosystems in the Galapagos have two seasonal periods. The humid, rainy period extends from December to May and has an average temperature of 29˚C. The dry, cold period extends from May to December with an average temperature of 22˚C [39]. Ninety-seven percent of the island areas are protected, and the remaining 3% contain human transformed ecosystems.

The island of Santa Cruz has the most inhabitants and receives the most tourist visits [47]. The main vegetation zone in Santa Cruz, along with the other islands, is the arid zone [40, 41] located between 0 and 300 m.a.s.l., characterized by an average temperature of 24˚C [39, 43]

The island of Santa Cruz comprises four vegetation zones: i) littoral, ii) arid, iii) transition, and iv) humid [43, 44]. Second in extent to the volcanic rock that covers most of the island, the arid zone covers 25% of its area [27]. The island has 25,244 inhabitants [48]. The main environmental issues in Santa Cruz are related to the economic industries associated with urbanization, agriculture, cattle ranching, and tourism [49]. Our study was conducted in the urban area of Puerto Ayora because this is located in the arid zone were most legume plants are distributed. This is an urban area with a very high population density of 450 inhabitants per hectare [50], that is subject to different anthropogenic activities, such as cattle ranching, urbanization, tourism, and the introduction of exotic legume species, which combine to produce large modifications in ecological interactions.

We sampled in the arid zone of Puerto Ayora in areas areas that comprise the following land uses: roadside (5 sites, trees from two species), which are linear areas on both sides of the roads, with trees 6 to 20 m tall; urbanization (two sites, trees from three species) located in the

port of the island, which includes a large area of planted tree species and managed or unmanaged forest; wooded pasture (three sites, trees from two species), which comprises abandoned land with areas of pasture land with trees, grasses and volcanic rock where invasive species predominate as a result of natural regeneration and dispersal processes; and plantation of *Leucaena leucocephala* (one site, one species of tree: the invasive *L. leucocephala*) corresponding to an area of approximately 15 × 15 m, located next to the Artisans' Park, where L. *leucocephala* trees of 6 to 12 m in height predominate; the trees are used by the inhabitants for timber extraction.

## Sampling and identification of plants and arthropods

The first task required to achieve our objective was to collect samples of seeds of the species of legume plants under study in areas under four different land uses. Sampling was carried out during the dry and cold season from August to September 2018. We located areas on the island in the arid zone and within the deciduous forest using a vegetation zone map for the archipelago [51]. We sampled within the arid zone because, after volcanic rock, it covers most of the island area. In addition, this area has a larger diversity of legume species and is the most affected by anthropogenic transformation. The areas sampled were located between 10 and 300 m.a.s.l. [43, 51]. We identified sites where legume trees had mature seedpods, and then we collected the mature seedpods according to their availability. For each sampled tree, we recorded the land use type, species, and geographical coordinates, resulting in samples from 10 geographical coordinates within the mentioned land uses. Land use type and the species collected from each site are shown in Table 1. The following exotic species were found: *L. leucocephala*, *L. trichodes*, *Senna bicapsularis*, *S. obtusifolia* and *Caesalpinia pulcherrima*. Plant identification was performed using specialized keys and the collaboration of botany experts from the "Herbario de la Fundación Charles Darwin" at Santa Cruz. Species and family names follow Tropicos.org [52].

**Table 1. Associations between arthropods and plants found on each land use type on the island of Santa Cruz, Galapagos, Ecuador.** Superscript "a" denotes species involved in the PSP scale of analysis; superscript "b" denotes the species involved in the PSPP analysis. RS: Roadside, WP: Wooded pasture, PL: Plantation of *L. leucocephala*, U: Urbanization.

| Plant species<br>Arthropod species | *L. leucocephala* | *L. trichodes* | *S. obtusifolia* | *S. bicapsularis* | *C. pulcherrima* |
|---|---|---|---|---|---|
| *Acanthoscelides machala* [a,b] | WP | - | RS | U | - |
| *Acanthoscelides macrophthalmus* [a,b] | RS | WP | - | - | - |
| *Acanthoscelides* nov. sp. [a, b] | RS, WP, PL | RS | - | - | U |
| Anthribidae sp. 1 [b] | RS | - | - | - | - |
| Cerambycidae sp. 1 [b] | PL | - | - | - | - |
| *P. pallidula* [b] | RS, PL | WP | - | - | - |
| Rhyzophagidae sp. 1 [b] | RS, WP, PL | WP | - | U | - |
| *Scutubruchus ceratioborus* [a,b] | - | U | - | - | - |
| *Crematogaster* sp. [b] | RS | WP | - | - | - |
| *Eupelmus pulchriceps* [a,b] | RS | - | - | - | - |
| *Horismenus* sp. [a,b] | RS, WP | WP | - | - | - |
| *Pteromalus* sp. [a,b] | RS, WP | - | - | - | - |
| Miridae sp. 1 [b] | RS, WP, PL | - | - | - | - |
| Nabidae sp. 1 [b] | RS, WP | - | - | - | - |
| Lepidoptera sp. 1 [b] | RS, WP, PL | WP | - | - | - |
| Myriapoda sp. 1 [b] | RS, PL | - | - | - | - |
| Aranae sp. 1 [b] | RS, PL | WP | - | - | U |

Seedpods from each individual tree were placed in plastic bags with label with the tree number, tree species, land use, and geographical coordinates. Once in the laboratory, the seedpods were split open, and all arthropods found in the seedpods and in the seeds were collected and preserved. We obtained an average 218 g of seeds from each of the tree species found in each land use type, which represents an average of 3920 seeds per species per land use. The seeds were placed in plastic containers kept at room temperature (at an average of 22.4˚C and 92% relative humidity). The seeds were inspected every 12 hours for 20 consecutive days to collect all emerging arthropods. We ceased examination of the seeds seven days after the last arthropod emerged.

Preserved arthropods were identified to the lowest taxonomic level possible using identification keys for Hymenoptera [53, 54], the expertise of a Hymenoptera taxonomist, identification keys for Coleoptera [55–58], the expertise of Bruchinae taxonomists, and identification keys for the remaining groups [59].

Permits to collect the samples were obtained from the Directorate of the National Park of Galapagos (DPNG permit: 7218). Permits to transport and export the insects to Colombia were obtained from the Agencia Nacional de Licencias Ambientales (ANLA permit: 01701). Permits to import the insects to Colombia were obtained from the Agencia Nacional de Licencias Ambientales (ANLA permit: 01605). No protected species were collected in this study. This study complies with the requirements of and was approved by the Comité de Investigación y Etica, Facultad de Estudios Ambientales y Rurales of Pontificia Universidad Javeriana-Bogotá, Colombia (minute 091 of April 27, 2018), and DPNG permit 7218 Sampling for this research was restricted to August-September 2018 due to the logistics involved with procuring permits to sample and export samples from the island of Santa Cruz.

## Data analysis

To determine the effects of land use and number of trophic levels on the structure of the interaction networks associated with the five legume species, interaction networks for each land use and networks with different number of trophic levels were constructed using adjacency matrices in Excel. The matrices included the abundances for each interaction between arthropods and plants. Networks composed of legume-seed beetle-Hymenoptera parasitoids are referred to hereafter as PSP, and networks with more than these three trophic levels composed of the PSP group plus predators and other phytophagous species are referred to hereafter as PSPP. The land uses compared were roadside, urbanization, wooded pasture, and plantations of *L. leucocephala*.

We first described the networks using three parameters: Connectance, Interaction evenness (IE), and Interaction strength asymmetry (ISA). We used these parameters to understand whether land use and number of trophic levels exert influence on the dynamics of interactions in the ecosystem involving insects and plants associated with the seeds of the five exotic legume species studied. We used connectance metrics to assess whether interaction networks exhibit greater or lesser complexity as a function of land use and the trophic structure present. In this context, greater connectance could indicate greater complexity in the network, which in turn is related to greater stability in the ecosystem, while lower connectance could denote a more fragmented or simplified network, which could be attributed to changes in land use or alterations in the trophic structure of the community.

We used the interaction evenness (IE) to assess whether some plant or insect species benefit disproportionately in these interactions, which could evidence the critical role of certain keystone species in shaping the network. Alterations in interaction equality may indicate how land use and the number of trophic levels affect the abundance or distribution of species in the network, thus providing valuable information on community structure.

Finally, we use the interaction strength asymmetry (ISA) to identify how insect plant interactions vary and to determine whether there is a dependence between them. This allows us to understand whether land use and trophic structure influence the very nature of interactions, which has important implications for the ecology and conservation of these ecosystems. i) Connectance measures the interactions found as a proportion of all possible interactions in the network [24]. This index determines the sensitivity of the community to environmental perturbations [60, 61]. Its values range from 0 to 1, where 0 is the lowest connectance and 1 is the highest [62]. (ii) IE measures the uniformity in the diversity of interactions in the network. This index varies from 0 to 1, where 0 indicates no uniformity of interaction and 1 is complete uniformity where all species have the same frequency of interaction [63]. iii) ISA is a measurement of specialization that explains the relative frequencies of insect and plant species. The index varies from -1 to 1, where a value of -1 indicates the lowest dependency of an animal on a plant and a value of 1 indicates the highest dependency of the plant on the animal has a value of 1 [63, 64]. To compute network-level metrics and plot the figure, we used the bipartite package [63]. To evaluate whether the observed values of the metrics were significantly different from random networks, we performed a comparison with null models using the t test of the bipartite package. Analyses were performed using R version 4.0.4. [65].

Second, we tested the effects of the number of trophic levels and land use on the parameters of the interaction networks. We first calculated the sampling intensity for each land use by using the sampling intensity index [66]. This index compares samples with different sampling efforts. In our case, we obtained seeds from different species, but the number of trees (and consequently of seeds) of each species was different. This estimate considers the need for more observations as the species richness increases [67], and it is calculated accounting the number of interactions, the number of species of plants and the number of species of animals. We evaluated differences in the sampling intensity with a chi-square test at a significance level of p ≤ 0.05. Differences among the metrics of the network in response to land use and number of trophic levels were calculated using a generalized linear model (GLM) at a significance level of p ≤ 0.05. Sampling intensity was used as a covariate because the parameters of the networks could vary in response to the size of the sample [67, 68]. The the *nlme* package was used for analysis [69]. Finally, we compared the observed values of the networks metrics to the expected values using a null t test that randomizes interactions while limiting the indicators that can be chosen [63].

## Results

We found 5552 arthropods belonging to six orders, 11 families and 17 species, forming 30 interactions (Table 1) out of a total of 47000 seeds collected from the five legume species. The most abundant species were the beetles *Acanthoscelides* nov. sp. (3274 individuals) and Rhizophagidae sp. (752 individuals). The least abundant species was also a beetle, *Scutobruchus ceratioborus* (1 individual). In the cases in which we were able to compare the observed values of the network metrics to the expected values under a null model, all values were highly significant, suggesting confidence in the obtained metrics (Table 2).

Hereafter, the results are described for each metric regarding the two factors evaluated: number of trophic levels, and land use. We did not find significant differences in the sampling intensity covariate, among land use types at any number of trophic levels (PSP: $X^2_3 = 0.035$, $p > 0.05$; PSPP: $X^2_3 = 0.038$, $p > 0.05$).

The number of species and interactions was highest on roadsides at the PSPP scale and lowest in plantations at the PSP scale (Fig 1). The number of species at the PSP scale was almost half that at the PSPP scale. Moreover, the number of interactions at the PSP scale was almost three times lower than that at the PSPP scale.

**Table 2. Parameters of the arthropod–exotic legume plant interaction networks on the island of Santa Cruz and their respective p values.** SI: Sampling intensity; C: Connectance; IE: Interaction evenness; ISA: Interaction strength asymmetry; IR: Interaction richness; AR: Arthropod species richness; PR: Plant species richness. *p* values refer to the effect of each land use type and number of trophic levels on each of the parameters estimated by GLM, using sampling intensity as a covariate.

| Land use type | Scale | SI | C | *p* | *IE* | *p* | ISA | *p* | IR | AR | PR |
|---|---|---|---|---|---|---|---|---|---|---|---|
| Roadside | PSP | 0.87 | 0.5 | 2.43e-06 | *0.3* | 3.28e-19 | -0.66 | 1.01e-07 | 6 | 6 | 2 |
| | PSPP | 0.94 | 0.5 | 2.04e-06 | *0.38* | 4.80e-18 | 0.86 | 1.92e-07 | 15 | 15 | 2 |
| Urbanization | PSP | 0.70 | 0.33 | 1.12e-07 | *0.24* | 3.89e-12 | 0 | NA | 3 | 3 | 3 |
| | PSPP | 0.79 | 0.33 | 2.38e-11 | *0.3* | 6.77e-16 | 0.51 | 6.47e-10 | 5 | 5 | 3 |
| Wooded pasture | PSP | 0.89 | 0.60 | 3.29e-08 | *0.39* | 2.40e-18 | 0.50 | 3.01e-07 | 4 | 3 | 2 |
| | PSPP | 1.04 | 0.63 | 2.09e-11 | *0.28* | 3.52e-22 | 0.83 | 6.20e-10 | 12 | 9 | 2 |
| Forest plantation | PSP | 0.70 | 1 | NA | *NA* | NA | 0 | NA | 1 | 1 | 1 |
| | PSPP | 0.94 | 1 | NA | *0.23* | NA | 0.75 | NA | 8 | 8 | 1 |

We found a significant effect of land use on connectance and IE, and no significant effect of the number of trophic levels on any of the parameters. Specifically, connectance was lower than 0.5 in most cases except plantation, which had a value of 1.0 for both networks of different trophic levels and in wooded pastures for the PSP network, which had a value of -0.66. The IE changed significantly in response to land use but not in response to the number of trophic levels. The sampling intensity covariate also had a significant effect on IE (F = 1460.8, df = 1, p = 0.01). Values close to one for all land use types, irrespective of the number of trophic levels, showed generalist networks. Given the lowest richness of the plantation land use, it did not make sense to estimate this index because *L. leucocephala* was the only associated plant species. ISA had a negative value for the roadside land use in the smallest PSP network. The remaining land use types in this network (PSP) had values of zero or very close to zero. In contrast, the land uses with a higher number of trophic levels (PSPP) exhibited positive values over 0.5 (Fig 1; Table 2).

## Discussion

Our study analyzed the effect of land use and number of trophic levels on the structure and complexity of the interaction networks of arthropods and plants associated with the seeds of five species of exotic leguminous trees in the arid zone of Santa Cruz Island, Galapagos archipelago. Parameters associated with the diversity of interactions, such as IE, are sensitive to community size (number of trophic levels in our study) and the frequency of interactions [70, 71]. We found no difference in these parameters among the networks when comparing PSP and PSPP. However, we did find differences in the networks with respect to land use. Several studies have demonstrated that community size, in our case represented by the number of trophic levels, directly affects the diversity of interactions [72–74]. Other studies [31, 33] found that forest fragmentation negatively affected the number of interactions compared to those in preserved and continuous forests. Thus, our results showed that the matrix (land use type in this case) in which the networks were immersed is more relevant than the number of trophic levels in the configuration of interactions among the community participants. In addition, when only the number of trophic levels was considered, the results were similar to those found in other studies; that is, the number of trophic levels directly affected the diversity of the interactions [70, 71]. However, when other parameters, such as land use, were included in the analysis, the effect of those parameters varied.

One of the more conspicuous plant species in the network deserves special mention: the invasive *L. leucocephala*. Plantation could represent original establishment of this species on the island of Santa Cruz about 35 years ago [49]. Because of this, when comparing the

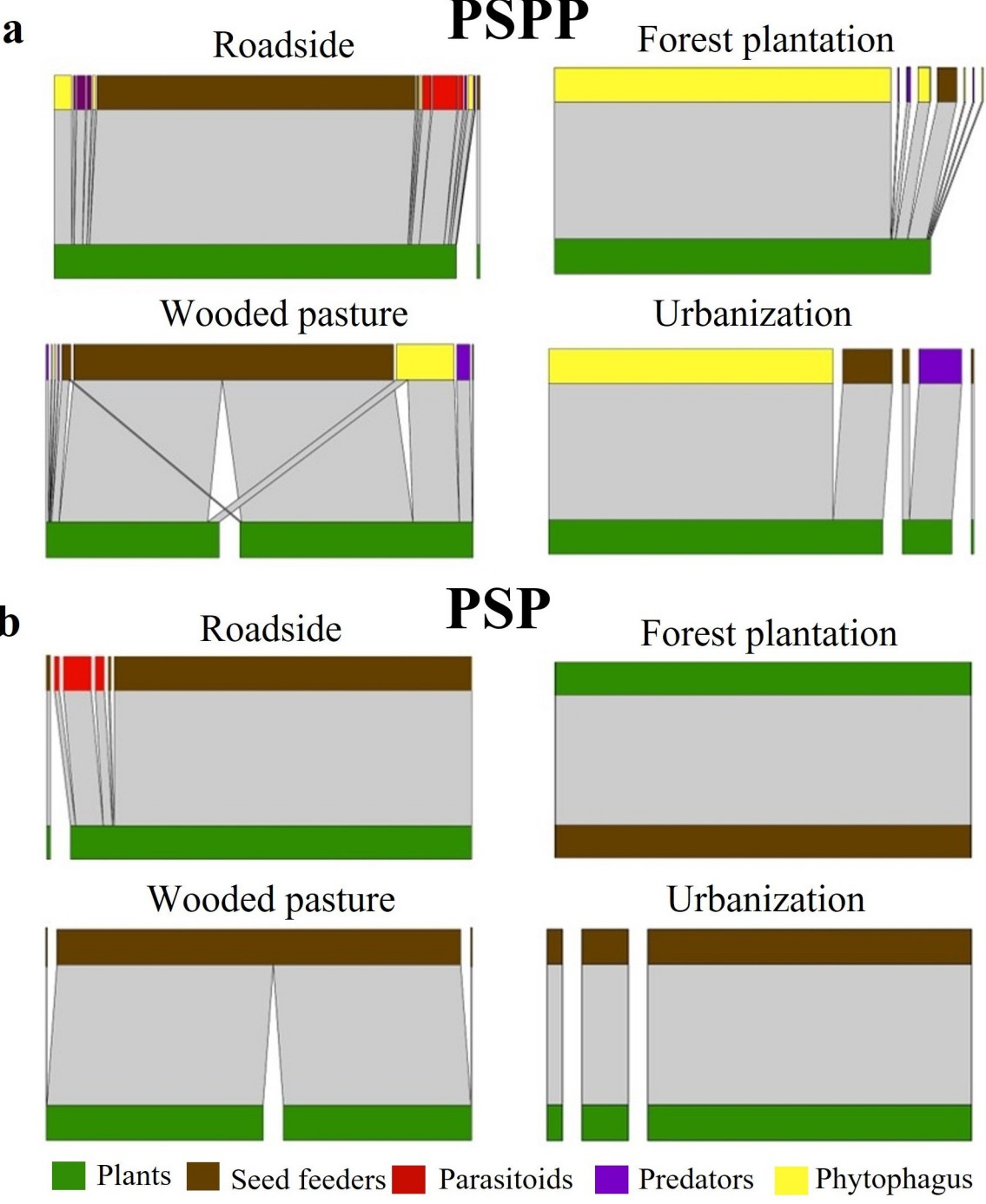

**Fig 1. Interaction networks between arthropods and exotic legume plants with two different numbers of trophic levels and on four land uses.** (a) PSPP: Legume–Seed beetle–Hymenoptera Parasitoids–Predators–Phytophagous; (b) PSP: Legume–Seed beetle–Hymenoptera Parasitoids. Lines connecting plants and arthropods represent observed interactions, and the broader the line is, the higher the frequency of the interaction.

networks of this land use with the others, it was evident that a more homogeneous community has reduced the number of interactions. The higher complexity of the networks in the remaining land use areas suggested that *L. leucocephala* may be driving resources away from native

species. Because it is an invasive species, it could also negatively affect local populations of native legumes [38] and even become a threat to their preservation on the island. Further studies comparing the interaction networks between *L. leucocephala*, other exotic species, and native legumes would allow us to test these assumptions.

Anthropogenic fragmentation of original ecosystems produces isolated patches of vegetation [20, 75]. Studies in perturbed ecosystems show that roadsides produce new environmental conditions, such as reduced humidity, higher temperature variation, and higher light penetration, among other conditions that may have positive or negative impacts on biodiversity [76–78]. In our case, we found that roadsides had more diverse species and interactions. We observed that in Puerto Ayora, most gardens and sidewalks were planted with ornamental herbs and trees other than legume species, which have lower aesthetic values in many cases. This may help to explain why the lowest diversity was found in the urban area.

Land use had a significant effect on connectance. In our study, the land use with the highest value (1.0) was the plantation because it only had one species of legume and one associated seed beetle. Some studies suggest that the higher the connectance of the network is, the higher its stability, which indicates a well-preserved ecosystem [60]. However, there is evidence to the contrary [79], and in our case, this high connectance value corresponded to the simplest and most transformed land use. Other studies [80] have shown that networks, depicting trophic interactions with a high connectance have fewer invasive species. Thus, we cannot interpret with confidence the significance of this metric to community structure [66]. However, one possible way to explain this metric is through a study that demonstrated that networks containing invasive species are more resilient to extinction events [81], suggesting that eradication of the invasive *L. leucocephala* may be more difficult in this land use than others because the interaction network is more stable. In addition, because our study included only exotic species of plants in the analysis, a comparison with networks associated with native legume plants is needed to clarify the role of native species in the more complex networks of insects and legumes on the island.

In conclusion, the most relevant factor responsible for the differences in the structure of arthropod communities associated with the five investigated legume species was land use type rather than the number of trophic levels involved in the analysis. The simultaneous analysis of these two factors enabled the identification of diverse and more complex patterns of community structure. In addition, we found the invasive species *L. leucocephala* to be the most conspicuous species of plant in the networks, which could represent a source of colonization toward the preserved dry forest in the Galapagos, as human transformation is increasing due to increased urbanization. The development of plans for the eradication or control of this invasive species should consider the links it has created with arthropods and the land uses in which it is more abundant. Additional research drawing comparisons among islands and among native and exotic species in the land uses we studied is recommended to understand and develop stronger conservation plans that consider the increased scientific evidence about effects of anthropogenic changes on ecosystems.

## Acknowledgments

We thank the Galapagos National Park Directorate (GNPD), Danny Rueda, Christian Sevilla, and park ranger Jibson Valle for use of the work facilities and field and logistic assistance. We also thank Charlotte Causton and Jacqueline Rodríguez, scientists from the Charles Darwin Foundation, who provided laboratory space for sorting seeds and rearing beetles. Entomology specialists Blanca Andrea Rodríguez and Geoffrey Morse provided valuable help with the identification of Hymenoptera parasitoids and some Bruchinae species, respectively.

## Author Contributions

**Conceptualization:** Angela R. Amarillo-Suárez, Mariana Camacho-Erazo.

**Data curation:** Angela R. Amarillo-Suárez, Mariana Camacho-Erazo.

**Formal analysis:** Angela R. Amarillo-Suárez, Mariana Camacho-Erazo.

**Funding acquisition:** Henri W. Herrera.

**Investigation:** Angela R. Amarillo-Suárez, Mariana Camacho-Erazo, Henri W. Herrera.

**Methodology:** Angela R. Amarillo-Suárez, Mariana Camacho-Erazo, Henri W. Herrera.

**Project administration:** Mariana Camacho-Erazo, Henri W. Herrera.

**Resources:** Henri W. Herrera.

**Supervision:** Angela R. Amarillo-Suárez, Henri W. Herrera.

**Writing – original draft:** Angela R. Amarillo-Suárez, Mariana Camacho-Erazo.

**Writing – review & editing:** Angela R. Amarillo-Suárez, Mariana Camacho-Erazo, Henri W. Herrera.

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
