## [Decision Letter · Decision Letter 0]

26 Dec 2022

PONE-D-22-17757Land use is more of a determinant than community size of ecological network complexity of arthropods associated with exotic legumes in the Galapagos, EcuadorPLOS ONE

Dear Dr. Suárez,

Thank you for submitting your manuscript to PLOS ONE. After careful consideration, we feel that it has merit but does not fully meet PLOS ONE’s publication criteria as it currently stands. Therefore, we invite you to submit a revised version of the manuscript that addresses the points raised during the review process.

First, I would like to apologize for the long delay in getting this paper returned. I contacted nearly 30 individuals to get the three reviews that are included. Those reviews vary greatly. However, they all indicate that at least some changes are needed. I suggest that you pay particular attention to the second and third reviewers who raised substantial concerns.  Please be sure to address them all in a revision or in the repose to reviewers.==============================

We look forward to receiving your revised manuscript.

Kind regards,

Sean Michael Prager, Ph.D.

Academic Editor

PLOS ONE

https://journals.plos.org/plosone/s/fileid=ba62/PLOSOne_formatting_sample_title_authors_affiliations.pdf.2.

“-HWR: IDIPI-104: Research Institute of the Escuela Superior Politécnica de Chimborazo (ESPOCH), Riobamba, Ecuador, https://www.espoch.edu.ec/ NO

ARAS: ID00008226; Pontificia Universidad Javeriana in Bogotá, Colombia. https://www.javeriana.edu.co/inicio NO”

Reviewers' comments:

Reviewer's Responses to Questions

**Comments to the Author**

1. Is the manuscript technically sound, and do the data support the conclusions?

Reviewer #1: Partly

Reviewer #2: No

Reviewer #3: Partly

2. Has the statistical analysis been performed appropriately and rigorously? 

Reviewer #1: Yes

Reviewer #2: No

Reviewer #3: Yes

3. Have the authors made all data underlying the findings in their manuscript fully available?

Reviewer #1: Yes

Reviewer #2: No

Reviewer #3: Yes

4. Is the manuscript presented in an intelligible fashion and written in standard English?

Reviewer #1: Yes

Reviewer #2: Yes

Reviewer #3: Yes

5. Review Comments to the Author

Reviewer #1: RE. Review of PONE-D-22-17757, “Land use is more of a determinant than community size of ecological network complexity of arthropods associated with exotic legumes in the Galapagos, Ecuador”

The manuscript, submitted to PLOS ONE for review, details an experiment in which legume seedpods were collected from five legume species found in Ecuador at sites where the land is used in different ways. The seedpods were held in laboratory conditions and arthropods using the seedpods were collected. The authors then conducted analyses to determine if the complexity of the communities associated with the seedpods (number of trophic levels, referred to as ‘community size’) or if the land use type where the seedpods were collected had a greater impact on the interactions within each community network. The research was conducted to fill knowledge gaps in the field of ecological network analysis, particularly with regard to interactions between insect herbivores and their natural enemies and the effects of human interference on interaction networks.

The authors clearly and concisely describe the current status of research on ecological networks in relation to human interference, especially invasive species. Thus, the Introduction provides ample justification for the work. Although some specific details are missing or are not clearly explained in the Methods (e.g., sampling and total number of sites sampled, please see specific comments below), the work done and the analyses conducted seem sound (although I am not an expert on these aspects of community ecology). The contribution(s) and findings of the work described in the manuscript are addressed in the Discussion and further areas of inquiry are highlighted. Overall, the work is of interest to a broad audience, spanning biologists, ecologists, entomologists, and likely policy analysts and will contribute to valuable discourse in this area of study. I support publishing the manuscript following revisions to clarify the experimental design and the presentation of some of the results.

Specific Comments by Line Number:

L24. The term “community size” is somewhat misleading. In general, community size is used to describe the number of individuals in a population and perhaps the geographic area occupied by a given community. Here, the authors use “community size” to refer to the number of trophic levels in the communities studied (tri-tropic vs. >3 trophic levels). Perhaps ‘trophic complexity’ or a similar descriptor could be used in place of “community size” throughout the manuscript, so that the factors being studied are more clear to the reader?

L33-35. Should this sentence read: “…and no significant effect of community size on connectance, diversity, or interaction evenness.”?

L40-46. This paragraph seems out of place in the Abstract and should be incorporated into the previous paragraph or deleted.

L119-120. The final sentence of the Introduction is a statement of Results, that may be better left to the Discussion.

L134. Why was sampling only conducted in one year? Would the authors expect to see differences between years that could impact the outcome of the study and the conclusions drawn from the analyses?

L135. Of the four vegetation zones on the island, were all of the deciduous forest areas within the humid zone? As the zones are defined in L125-126, please define which zones the samples were collected in.

L136-138. How many sites of each land use type were sampled and how many sites were sampled in total? Ten total sites divided by 4 land use types is an inadequate number of sites for rigorous conclusions to be drawn from. Please clarify this aspect of the experimental design.

L138-141. Why collect seeds from five legume species and not just focus on one legume species (or on one endemic species and one invasive species)? Were all five species available at all of the sites that were sampled? Could differences in plant diversity at the sites have impacted the analyses and subsequent conclusions of the study? Sampling intensity is used as a covariate and does not appear to have had a major effect, but plant diversity effects are not directly addressed. Please clarify these aspects of the experimental design.

L173. Use “Chi-square”

L213-223. With regard to formatting, the paragraph beginning on L213 and the paragraph beginning on L223 should be melded into one paragraph, with the Figure legend/title and Table 2 appearing at the end of the text for that new paragraph. This issue arises from the formatting requirements of the journal and is not the fault of the authors, but does make the text of the Results difficult to follow.

L215-222. In the title for Figure 1, I believe the colour blue correspond to arthropods that eat plants (but not plant seeds). If this is correct, then ‘herbivore’ would be a better descriptor than ‘phytophagous.’ The final sentence in the figure title could be revised to read: “Lines connecting plants and arthropods represent observed interactions, where broader lines represent interactions that occur with high frequency.”

L223-224. Table 2 provides a nice summary of the species found in the study and the land use areas where they were found, however, it does not classify or indicate which species were found in PSP communities and which were found in PSPP communities as this sentence indicates. Please revise to ensure the Table is being used and referred to accurately.

L227. Is the F-value correct? Please double check.

References (starting L307): The manuscript and concepts therein have been thoroughly researched. However, the authors cite 87 sources, which is quite excessive for a research paper. If possible, the number of references should be reduced.

Abbreviations are not used in excess in the manuscript, but some abbreviations used in the text are not necessary. For example, the terms ‘interaction evenness’ and ‘interaction strength asymmetry’ could be used in the text rather than the abbreviations IE and ISA, respectively. Doing so would help readers to more easily keep track of which ecological measures are being analyzed, especially readers that are more familiar with the arthropod communities than with these ecological concepts.

Reviewer #2: The authors collected 250 grams of seeds from 5 legume tree species at 10 sites that represented 4 land-use types from within the arid zone of Santa Cruz Island, Galapagos, over 2 months in 2018 and reared the associated insects. A number of community metrics were calculated at two community scales, one including all plant and insect species and one including all species except predators. The effect of land-use types on these parameters was assessed using GLM.

One of my first impressions of the article was that it is based on very little sampling over a very short time period compared to other similar studies. However, I have to admit that very many insect individuals were reared, so maybe there is sufficient power to detect some trends.

Some of the main results of the study were a series of community indices, but there seems to be no basis upon which to decide whether the values of these indices are ‘significant’ or what 95% confidence limits are for them. I presume that indices such as these could be calculated for even a very few samples so it is hard to evaluate how seriously to take them. I would think there is some way to estimate confidence, for instance by the comparison to a null value through a bootstrapping process. Approaches like this are available through the ‘econullnetr’ R package.

Other results concern trends associated with the landscape types. I found these to be frustrating though since there was no attempt to correct these effects for host plant species. I would think that host plant is an important driver of insect diversity in a study like this, and while it was clear that the composition of the host plants varied among the site categories from the data tables, there was no quantification of which species predominated at which types of sites. It is not hard to imagine that the trends being ascribed to ‘land-use’ had alot (or at least something) to do with the host-plants present at these sites. I think that an analysis taking this into account would be necessary to properly interpret the results.

Related to this problem is that this paper is almost completely devoid of natural history. A reader would have no idea which herbivore species is feeding on which plant species, which parasitoid species is feeding on which herbivore species or which predators are present on which plants. Some of the linkages can be figured out from the tables but not most. I really miss these kinds of details and think it is a lost opportunity to ignore them; without them the paper feels very dry and technical.

So – to summarize I feel that the paper as it stands feels like a preliminary study from a limited dataset and provides analyses that are very hard to have confidence in.

I have some more minor comments below:

Abstract: the abstract is too long and should not contain statistical details.

L40. There seems to be some repeating of material here.

L97. It would be good to explain the difference between exotic and invasive here

L112. I think you mean ‘dispersal’ not ‘dispersion’

L134. ‘season’ not ‘seasons’

L137. Saying ‘including’ in this context is vague. You must list all of the landscape types sampled, along with a description of each and an indication of how many sites of each. From what is presented it looks like there were 4 site classifications and 10 sites so presumably 2 or 3 sites of each landscape type (which seems like very low replication) but it was not clear enough.

L155. ‘Directorate’ not ‘Direction’

L164-166. I don’t understand the rationale between using two ‘community sizes’ for the network analysis (one with predators and one without). To me it just makes things more complicated.

L168. Please indicate which R package was used.

Fig. 1 would be more useful if the species involved could be indicated.

L256-7. I don’t understand this sentence; the grammar needs to be fixed I think. I assume that the authors do not mean to say that plantations represent the ancestral state of Santa Cruz Island, but that is what the sentence seems to say.

L265-6. I don’t understand this sentence either. What are ‘island-like patterns’? I would think the island was more ‘island-like’ before the arriv

Reviewer #3: The manuscript seeks to examine the effects of land use and community size on a plant-herbivore network system, which is a very interesting and overlooked topic in community and plant-animal interaction ecology. The importance of the study has been well contextualized and placed compared to other studies in the field. The narrative and the study design are excellent and straightforward, but the rationale needs major enhancements for more cohesion throughout the manuscript. Specifically, the research question and the objectives are not clearly stated making the story they want to highlight a little bit blurry. Additionally, tying those objectives with the methods and results needs more work as I am not sure what activities have been done to address which objectives. The authors may also want to correct the discussion accordingly.

L1 - Title. Could be shortened to “Land use is more of a determinant than community size of ecological network complexity”.

L4 - Short title. Typo on “networks”; change “on” to “with”.

Abstract. The implications of your results are missing: we have the results, but what they are supposed to mean? I suggest the authors connect the results directly with their interpretation and/or implications.

L23: variation of what?

L25-27: Is the whole network comprised of exotic plant species or is there only five exotic species in the whole network system?

L25: are seed beetles and Bruchinae the same taxon? If so, make sure to make that clear here as we are not all insect specialists to connect the dots between them.

L26: what are the four land use types?

L33-46: I suggest removing or reducing statistical reports to the minimum. I suggest focusing on the biological meaning of those statistical significances and the technical terms (connectance and IE) in the abstract to highlight your results and their implications and make it more accessible for non-network specialist scientists.

L40-46: why is the last paragraph separated from the whole abstract?

L46. is this sentence complete or is there some explanation to it?

Keywords. Missing

Introduction. Excellent outline and connection of the main concepts of the studies. However, there are too many repetitions of the same information. I suggest that the authors go over the manuscript and make sure one piece of information is stated once and does not repeat it again in another paragraph. As I mentioned earlier, the research question and the objectives of this manuscript are not clear while its importance is well-outlined.

L52: a consensus or evidence? Or is there any missing information?

L54: remove abundance and richness; they are already comprised in functional and taxonomic biodiversity.

L53-55: isn’t the loss of species that creates biodiversity functional loss? You seem to argue the opposite here.

L57-58: remove “in addition to other factors”.

L62: can you advance one or two reasons why the effects of anthropogenic activities on plant-herbivore system have been disregarded despite their importance in the ecosystem?

L65-66: this sentence needs a little more explanation.

L67: is there a gap of knowledge on the ecology of plant-herbivore and parasitoids, or on the effects of anthropogenic activities on this system? If both are lacking, I suggest expanding more on both to highlight the gap and the importance of addressing it.

L68-70: As this is an introductory sentence, maybe add another reference that is not focused on plant-herbivore network, such as plant-pollinator or plant-seed disperser network.

L70-72: not clear.

L74-76: maybe synthetize reference 13, 17, 23-25 into one sentence or two to show the change in ecological networks in response to the introduction of exotic species instead of just citing them.

L77: cite both the original reference 26 and its erratum.

L77-78: this is true, but how?

L779-89: why do you come back to the impact of anthropogenic activities? I suggest this paragraph focusing on plant-herbivore network, what is known, and what is lacking, then introducing the lack of investigation in this field and the contradictions you found in the existing studies. Also, make sure to introduce the meaning of specific terms such as connectance. Finally, reorganize the ideas to avoid the back and forth between mutualistic networks and plant-herbivore networks.

L90-91: any interaction networks? Or just plant-herbivore networks?

L99-113: please move the information about the study site, study species in the Materials and Methods, and replace this last section with the goal and objectives of the manuscript as well as the study system you investigate. Maybe consider also explaining why Legumes are a big deal in this ecosystem.

L113-116: can you explain briefly how and add relevant references? Also, make sure to link this information about natural enemies in the rationale of your study design.

L116-120: please reorganize those sentences with the goals, objectives, study system above. Also, make sure to highlight why you specifically chose the arid zone among the other zones you mention in the study site section. Finally, I suggest you add a sentence or two to explain why you chose to examine two types of network system PSP and PSPP.

Materials and Methods. It is a very concise section although some key points are missing. However, I don’t think that is appropriate for such a technical manuscript. I suggest adding more details on some aspects of the methods, especially the data analysis (detailed below) and restructure the process following the research goal and objectives.

L126-127: not clear and cite only the relevant references.

L128-131: why is this information relevant here? If it denotes the different ecosystem of sampling sites, please say so.

L134: sampling of what?

L135-136: I am not sure I understand whether you sampled in the vegetation zone or the arid zone, or these two are the same.

L136: repetitive information.

L136-138: in how many sites did you sample exactly, five or ten? You say ten but only listed five.

L138-141: is there any particular reason to choose only these five species?

L146-147: 250 g of seeds from which species?

L148: please define the acronyms before using them. Also, check other acronyms throughout the text. Are these conditions standardized or is there any reason to choose them?

L153: delete the “,” between Coleoptera and Bruchinae. I suggest mentioning at the beginning of this section that you sample for those insects specifically as I am not sure if you sample for them specifically or for all arthropods earlier and then you only use identification keys for the Bruchinae here.

L161: Data analysis. Please consider reorganizing the process of analysis. There are too many “first steps, I am not sure which one goes with which one. Also, I suggest to start with the construction of the network, then the process before the actual analyses.

L162-164: Networks of what? Also, how did you characterize the networks from the sampling?

L166-168: What kind of variation? Also, do these land use types correspond to the number of sampling sites you mention earlier?

L170: Change to “This index compares samples…”

L172-174: It is a bit unclear, please consider revising the sentences. Also, did you use the index for your analysis? If so, please state so. If not, please explain better why this step is critical to your data analysis.

L178-180: Change to “This index determines the sensitivity…”

L190: Please explain the acronym before using it. Additionally, please make sure to mention the predictors and covariates, how many models did you run, and are there any other steps you took during the comparisons, if any.

L192: You can delete this sentence.

Results. Again, good job on the succinct writing composition! Based on the research goal and objectives here, this section fairly addresses them. I suggest revising the section by adding more structure and make the ideas better connected to fully answer the questions. Also, please do not repeat the results in the table in the text, and consider spacing the captions from the main text.

L196-198: are these species plant or an insect species?

L198-200: Was the sampling intensity applied to both plant and insect species or plant species only? And between which variables was it compared to (sites, etc.)? Please explain these more in the data analysis section.

L200-202: Please write the full value of P if > 0.05. Also, correct throughout the text.

Table 1: Please check to number of sampling sites throughout the text and make sure it is always the same. Sometimes, it is ten, sometimes five, and sometimes eight. Add the caption to the table right after the title and not after the table.

L211: Delete the repeated “PSPP”.

Figure 1: Also, please add an annotation of these indications directly on the figure (L218-221).

Table 2: Is there any way for this table to be combined with Fig. 1? It is a bit confusing as it is now and you can color-code each insect order for instance.

L227-241: can’t the stats be added to Table 1 or another table? It becomes daunting to read. Also, please consider revising the result report. I suggest starting with the PSP community first and report all results related to it, and then the same with the PSPP. Maybe also add some basic interpretation of the comparisons for each index here as for now it looks like a repetition of what the stats show.

Discussion. It is a sound and thorough discussion. Similar to other section, please reorganize the ideas to reflect more the research goal and objectives and the results report. Also, it becomes a bit unclear if it is the invasive species or the land use that really threatens Santa Cruz biodiversity in the context of this manuscript. It is ok to have them both here and it is well-thought, but consider restructure the arguments so you avoid repetitions and have better implications on the negative impacts of anthropogenic activities.

L247-248: you did or you didn’t?

L248-252: please revise the connection between these two sentences.

L252-254: please make sure to recall your hypotheses/predictions or research questions before saying the results turned out the be as expected.

L256: Change to “forest plantations could represent…”

L256-264: to which results this section corresponds to? Please add this before discussing the land use and your proposed hypotheses.

L272: Change “In our case” by “In addition”.

L272-273: Change to “roadsides had more diverse species and interactions”.

L273-275. This needs references, or mention it if it is a person observation.

L280-282: This is repetitive.

L283: Ok, so what alternative do you propose?

L284-285: and why is that?

L292: I suggest you add a paragraph talking about the conservation implications of your findings and/or the importance of addressing the plant-herbivore systems as you pointed out the lack of studies in this field and its importance for the ecosystem.

6. PLOS authors have the option to publish the peer review history of their article (what does this mean?). If published, this will include your full peer review and any attached files.

Reviewer #1: No

Reviewer #2: No

Reviewer #3: No

---

## [Author Response · Author response to Decision Letter 0]

21 Jun 2023

Response to reviewers:

Thank you very much for all comments made by the reviewers. They helped us to improve the quality and clarity of the manuscript. Our responses are in normal letter following each of the comments highlighted in bold type lettering. Line numbers in the responses correspond to lines in the “Manuscript” file.

https://journals.plos.org/plosone/s/fileid=ba62/PLOSOne_formatting_sample_title_authors_affiliations.pdf.2.

“-HWR: IDIPI-104: Research Institute of the Escuela Superior Politécnica de Chimborazo (ESPOCH), Riobamba, Ecuador, https://www.espoch.edu.ec/ NO

ARAS: ID00008226; Pontificia Universidad Javeriana in Bogotá, Colombia. https://www.javeriana.edu.co/inicio NO”

Response: The abstract in the online submission form has been corrected, and it is identical to the abstract in the manuscript.

Response: We completed the ethics statement in the Methods section, lines 180-187

Response: We have not included supporting information in this revised manuscript

Reviewers' comments:

Reviewer's Responses to Questions

Comments to the Author

1. Is the manuscript technically sound, and do the data support the conclusions?

Reviewer #1: Partly

Reviewer #2: No

Reviewer #3: Partly

2. Has the statistical analysis been performed appropriately and rigorously? 

Reviewer #1: Yes

Reviewer #2: No

Reviewer #3: Yes

3. Have the authors made all data underlying the findings in their manuscript fully available?

Reviewer #1: Yes

Reviewer #2: No

Reviewer #3: Yes

4. Is the manuscript presented in an intelligible fashion and written in standard English?

Reviewer #1: Yes

Reviewer #2: Yes

Reviewer #3: Yes

5. Review Comments to the Author

Reviewer #1: RE. Review of PONE-D-22-17757, “Land use is more of a determinant than community size of ecological network complexity of arthropods associated with exotic legumes in the Galapagos, Ecuador”

The manuscript, submitted to PLOS ONE for review, details an experiment in which legume seedpods were collected from five legume species found in Ecuador at sites where the land is used in different ways. The seedpods were held in laboratory conditions and arthropods using the seedpods were collected. The authors then conducted analyses to determine if the complexity of the communities associated with the seedpods (number of trophic levels, referred to as ‘community size’) or if the land use type where the seedpods were collected had a greater impact on the interactions within each community network. The research was conducted to fill knowledge gaps in the field of ecological network analysis, particularly with regard to interactions between insect herbivores and their natural enemies and the effects of human interference on interaction networks.

The authors clearly and concisely describe the current status of research on ecological networks in relation to human interference, especially invasive species. Thus, the Introduction provides ample justification for the work. Although some specific details are missing or are not clearly explained in the Methods (e.g., sampling and total number of sites sampled, please see specific comments below), the work done and the analyses conducted seem sound (although I am not an expert on these aspects of community ecology). The contribution(s) and findings of the work described in the manuscript are addressed in the Discussion and further areas of inquiry are highlighted. Overall, the work is of interest to a broad audience, spanning biologists, ecologists, entomologists, and likely policy analysts and will contribute to valuable discourse in this area of study. I support publishing the manuscript following revisions to clarify the experimental design and the presentation of some of the results.

Response: We appreciate very much the comments made by reviewer. They helped us to produce a better and more consistent manuscript. We addressed all comments, and the responses below explain the changes we made. We read the comments made by the three reviewers and revised some sections of the manuscript. Specifically, we made changes to the Experimental Design and Results sections to improve the consistency among different sections of the manuscript, as was also suggested by Reviewer #3.

Below are our responses to the comments of Reviewers

Reviewer #1:

Specific Comments by Line Number:

L24. The term “community size” is somewhat misleading. In general, community size is used to describe the number of individuals in a population and perhaps the geographic area occupied by a given community. Here, the authors use “community size” to refer to the number of trophic levels in the communities studied (tri-tropic vs. >3 trophic levels). Perhaps ‘trophic complexity’ or a similar descriptor could be used in place of “community size” throughout the manuscript, so that the factors being studied are more clear to the reader?

Response: Thank you very much for the comment. We considered this term and agree that it could be misleading to the readers. Therefore, we changed “community size” to “number of trophic levels”. We made this change throughout the manuscript. It is closer to what we intended to express.

L33-35. Should this sentence read: “…and no significant effect of community size on connectance, diversity, or interaction evenness.”?

Response: Yes, you are right. Thank you very much. We changed the sentence by eliminating the parenthesis. 

L40-46. This paragraph seems out of place in the Abstract and should be incorporated into the previous paragraph or deleted.

Response: Thank you very much again. We deleted the paragraph because it was not necessary.

L119-120. The final sentence of the Introduction is a statement of Results, that may be better left to the Discussion.

Response: Thank you very much. We deleted the sentence

L134. Why was sampling only conducted in one year? Would the authors expect to see differences between years that could impact the outcome of the study and the conclusions drawn from the analyses?

Response: Yes, it was. As stated the Methods section, the sampling for this study was conducted at a specific time and in a certain geographical area. The sampling was performed during the year the project was founded; consequently, legal permits were issued for only that period and sampling site. Our research was conducted in a biosphere reserve and a natural park in Ecuador where permits for collecting biodiversity samples are very strict and specify the time and space of sampling. It has been shown that interaction networks change over time, even on very small scales, such as in a matter of days (Bascompte, 2014, and references there in). Thus, it would have been bold, to say the least, to extend our conclusion to other times throughout the year. In addition, this was not the main objective of our study. We did not intend to make comparisons or to show how the interaction networks change over time. This could be the topic of a future study. We understand that it would be desirable for the conclusions of a study to have broader implications; however, they need to reflect the research and be consistent with the data collected to address the study aims. Our research question explored the effect of land use and the number of trophic levels in the interactions; our conclusions can be extended to similar settings but not to comparisons over time.

L135. Of the four vegetation zones on the island, were all of the deciduous forest areas within the humid zone? As the zones are defined in L125-126, please define which zones the samples were collected in.

Response: Thank you for the question. As mentioned in the manuscript, the study was conducted in the arid zone. Based on the reviewer’s question, we reviewed the paragraph and found it was unclear, so we changed some sentences in this section.

L136-138. How many sites of each land use type were sampled and how many sites were sampled in total? Ten total sites divided by 4 land use types is an inadequate number of sites for rigorous conclusions to be drawn from. Please clarify this aspect of the experimental design.

Response: We revised the Methods section. We hope it is now clear.

L138-141. Why collect seeds from five legume species and not just focus on one legume species (or on one endemic species and one invasive species)? Were all five species available at all of the sites that were sampled? Could differences in plant diversity at the sites have impacted the analyses and subsequent conclusions of the study? Sampling intensity is used as a covariate and does not appear to have had a major effect, but plant diversity effects are not directly addressed. Please clarify these aspects of the experimental design.

Responses: 

First question Why collect seeds from five legume species and not just focus on one legume species (or on one endemic species and one invasive species)? 

Indeed, the questions the reviewer ask would be very interesting to solve on other studies such as those the reviewer states: comparing communities associated to an exotic and a native species, even comparing the same communities among several years, islands, etc. However, that was not the focus of this study. Communities are structured according with their history of change and as we wrote before, that was not the focus of our study. We did what we did because the richness, diversity and interactions of communities of insects associated to a plant are not just depending of a specific plant and its insects. This specific interaction also depends of factors such as other plants that are used by the same insects and of the abundance of the different hosts in a specific land use, if the plant is in addition to exotic, invasive, etc. Exotic and invasive species respond different to community assemblage. In our study there was an invasive species plant (L. leucocephala), and the other species are considered as exotic (not invasive species) in the archipelago. 

Second question: Were all five species available at all of the sites that were sampled? 

Response: Table 2, figure 1 show this information. All five species were not all in the sampling sites. In order to gain clarity, we added a sentence and a citation to the table 1 in the methods section (Lines 152-162). We also modified table 1 to show the interactions among animals, plants, and the land use they were found.

Third question: Could differences in plant diversity at the sites have impacted the analyses and subsequent conclusions of the study? Sampling intensity is used as a covariate and does not appear to have had a major effect, but plant diversity effects are not directly addressed. Please clarify these aspects of the experimental design.

Response: In fact, it could. That is why we used the sampling intensity index to account for the variation in plants and animals’ richness in our analyses. The sampling intensity index is calculated having in account the number of interactions (abundance of interactions), the number of species of plants (diversity) and the number of species of animals (diversity), and it was calculated for each land use. That is how the metrics works. Lines 213-218 from the data analysis section explained this. However, in order to gain more clarity, we added some additional information in this section (lines 213-218). On the other hand, as much as we say plant diversity could have an impact, insect diversity could too.

L173. Use “Chi-square”

Response: Response: We modified the text, as suggested.

L213-223. With regard to formatting, the paragraph beginning on L213 and the paragraph beginning on L223 should be melded into one paragraph, with the Figure legend/title and Table 2 appearing at the end of the text for that new paragraph. This issue arises from the formatting requirements of the journal and is not the fault of the authors, but does make the text of the Results difficult to follow.

Response: We changed accordingly. We could changed back if the formatting requirements of the journal say so.

L215-222. In the title for Figure 1, I believe the colour blue correspond to arthropods that eat plants (but not plant seeds). If this is correct, then ‘herbivore’ would be a better descriptor than ‘phytophagous.’ The final sentence in the figure title could be revised to read: “Lines connecting plants and arthropods represent observed interactions, where broader lines represent interactions that occur with high frequency.”

Response: Phythophagus is a synonym of herbivorous, so we left the term “phytophagous”. We did not change the final sentence of the explanation of the figure as suggested by the reviewer because it does not clearly describe the connections. We left the description as it was previously.

L223-224. Table 2 provides a nice summary of the species found in the study and the land use areas where they were found, however, it does not classify or indicate which species were found in PSP communities and which were found in PSPP communities as this sentence indicates. Please revise to ensure the Table is being used and referred to accurately.

Response: Thank you very much. We changed Table 1 to clarify the interactions between the species on the land use areas and for different number of trophic levels. We also improved the accuracy of the description of the table. Table 2 is no longer necessary.

L227. Is the F-value correct? Please double check.

Response: Thank you very much. We changed from, 34,5 to 34.5.

References (starting L307): The manuscript and concepts therein have been thoroughly researched. However, the authors cite 87 sources, which is quite excessive for a research paper. If possible, the number of references should be reduced.

Response: We revised the references in the manuscript and retained those we believe are necessary

Abbreviations are not used in excess in the manuscript, but some abbreviations used in the text are not necessary. For example, the terms ‘interaction evenness’ and ‘interaction strength asymmetry’ could be used in the text rather than the abbreviations IE and ISA, respectively. Doing so would help readers to more easily keep track of which ecological measures are being analyzed, especially readers that are more familiar with the arthropod communities than with these ecological concepts.

Response: We made changes in some sections

Reviewer #2: 

The authors collected 250 grams of seeds from 5 legume tree species at 10 sites that represented 4 land-use types from within the arid zone of Santa Cruz Island, Galapagos, over 2 months in 2018 and reared the associated insects. A number of community metrics were calculated at two community scales, one including all plant and insect species and one including all species except predators. The effect of land-use types on these parameters was assessed using GLM.

One of my first impressions of the article was that it is based on very little sampling over a very short time period compared to other similar studies. However, I have to admit that very many insect individuals were reared, so maybe there is sufficient power to detect some trends.

Response: We modified the Methods section to make it clearer. 

Some of the main results of the study were a series of community indices, but there seems to be no basis upon which to decide whether the values of these indices are ‘significant’ or what 95% confidence limits are for them. I presume that indices such as these could be calculated for even a very few samples so it is hard to evaluate how seriously to take them. I would think there is some way to estimate confidence, for instance by the comparison to a null value through a bootstrapping process. Approaches like this are available through the ‘econullnetr’ R package.

Response: Thank you very much for your comments. We performed significance tests of the metrics, contrasting the observed values with those of the null models and found that all of them were significant. We added this information in the manuscript in the Methods section (lines 209-211) and the Results section (lines 233-235, table 2)

Other results concern trends associated with the landscape types. I found these to be frustrating though since there was no attempt to correct these effects for host plant species. I would think that host plant is an important driver of insect diversity in a study like this, and while it was clear that the composition of the host plants varied among the site categories from the data tables, there was no quantification of which species predominated at which types of sites. It is not hard to imagine that the trends being ascribed to ‘land-use’ had alot (or at least something) to do with the host-plants present at these sites. I think that an analysis taking this into account would be necessary to properly interpret the results.

Response: Thank you for the comment. First, we differentiated land use types, not landscape types. This has been consistently stated throughout the manuscript. Land use and landscape are different aspects of land characterization. Second, as stated in the Methods section, we used the sampling intensity index, which accounts for the diversity of plants, animals, and interactions in the algorithm, thus controlling (or correcting) for the variation in plants and animals as well. We agree with the reviewer’s view of the bottom-up regulation of communities in which plants determine the richness of herbivores and higher trophic levels. However, from a top-down perspective, natural enemies and herbivores also play an important role in structuring communities.

Related to this problem is that this paper is almost completely devoid of natural history. A reader would have no idea which herbivore species is feeding on which plant species, which parasitoid species is feeding on which herbivore species or which predators are present on which plants. Some of the linkages can be figured out from the tables but not most. I really miss these kinds of details and think it is a lost opportunity to ignore them; without them the paper feels very dry and technical.

Response: Response: Thank you for the comments. We don’t think this is a “related problem”. The previous comment was about how to control for variation in plant diversity and this comment suggests adding information regarding which species are related to each other. To include more natural history, we redesigned Table 1 to show the plant-insect interactions as well. We do not see a problem with the manuscript having a dry and technical tone. We apologize if this was frustrating. We hope the information we added meets the requirements for adding the natural history of the species involved in the networks.

So – to summarize I feel that the paper as it stands feels like a preliminary study from a limited dataset and provides analyses that are very hard to have confidence in.

Response: To summarize, we responded to every comment before and provided evidence by using the appropriate metrics and concepts to describe the pattern we found. We considered the comments of all three reviewers to improve the clarity of the manuscript. We hope this addresses the concerns of the reviewer, and that way he/she does not feel frustrated again.

I have some more minor comments below:

Abstract: the abstract is too long and should not contain statistical details.

Response: The abstract is within the word count established by the journal

L40. There seems to be some repeating of material here.

Response: Thank you for your comment. In deed the paragraph was repeated, so we deleted it. 

L97. It would be good to explain the difference between exotic and invasive here

Response: We did not find it necessary, nor did the other two reviewers. Therefore, we did not change this sentence.

L112. I think you mean ‘dispersal’ not ‘dispersion’: 

Response: We mean dispersion: “The spatial distribution of individual organisms within a local population” Dispersal is “the movement of organisms away from their point of origin (Lomolino et al., 2006). Or according to Nathan (2001, Encyclopedia of biodiversity) dispersion “refers to the fine-scale spatial distribution pattern of organisms” and dispersal is “the movements of disseminules away from their parent source”. The references cited support the idea of dispersion, not dispersal, which is a biogeographical process on a larger scale.

 L134. ‘season’ not ‘seasons’:

Response. You are right. We corrected the word.

L137. Saying ‘including’ in this context is vague. You must list all of the landscape types sampled, along with a description of each and an indication of how many sites of each. From what is presented it looks like there were 4 site classifications and 10 sites so presumably 2 or 3 sites of each landscape type (which seems like very low replication) but it was not clear enough.

Response: You are right. It is not “including”. Those land use types were the types that were sampled. We corrected the paragraph to make it clearer and added a description of each one (See lines 147 and 151).

L155. ‘Directorate’ not ‘Direction’

Response: Thank you very much. We implemented the suggested change.

L164-166. I don’t understand the rationale between using two ‘community sizes’ for the network analysis (one with predators and one without). To me it just makes things more complicated.

Response: Response: This study evaluated the effect of the number of trophic levels involved in the interactions and of the land use (not landscape) on the structure of the networks. To simplify things, we could have written a paper discussing just the interactions. However, our approach went beyond the description of the interactions. The ecology of communities changes as the number of interactions varies and as anthropogenic factors, such as land use, intervene. However, as we wrote in the Introduction, the effects of these factors do not exhibit the same trends. In addition, as Reviewer #1 stated, “The research was conducted to fill knowledge gaps in the field of ecological network analysis, particularly with regard to interactions between insect herbivores and their natural enemies and the effects of human interference on interaction networks. The authors clearly and concisely describe the current status of research on ecological networks in relation to human interference, especially invasive species. Thus, the Introduction provides ample justification for the work.” In addition, as reviewer #3 wrote, “The manuscript seeks to examine the effects of land use and community size on a plant-herbivore network system, which is a very interesting and overlooked topic in community and plant-animal interaction ecology. The importance of the study has been well contextualized and placed compared to other studies in the field.”

Consequently, we did not modify the description of the rationale of our study in the Introduction.

L168. Please indicate which R package was used.

Response: This has been done.

Fig. 1 would be more useful if the species involved could be indicated.

Response: Yes, it would be more useful. However, there is not enough space to add the names of the species to the figure. Most published papers do not include text like that because the figures get very messy. In addition, Table 1 shows which species interact with other species.

L256-7. I don’t understand this sentence; the grammar needs to be fixed I think. I assume that the authors do not mean to say that plantations represent the ancestral state of Santa Cruz Island, but that is what the sentence seems to say.

Response: We modified the phrase for clarity. 

L265-6. I don’t understand this sentence either. What are ‘island-like patterns’? I would think the island was more ‘island-like’ before the arriv

Response: The phrase “Anthropogenic fragmentation of original ecosystems produce vegetation cover with island-like patterns” refers to the fact that the fragmentation of ecosystems produces patches (islands) of vegetation. This is a basic theory of fragmentation, and it is written well, in correct English. However, here is an expanded explanation: This means that the vegetation in each patch is affected by environmental factors, as occurs on real islands (see references cited afterward phrase). As stated, it refers to the effects of fragmentation in the cover of vegetation. The subject of the sentence is not the island. We believe this is clear and disagree with the reviewer’s comment.

Reviewer #3: 

The manuscript seeks to examine the effects of land use and community size on a plant-herbivore network system, which is a very interesting and overlooked topic in community and plant-animal interaction ecology. The importance of the study has been well contextualized and placed compared to other studies in the field. The narrative and the study design are excellent and straightforward, but the rationale needs major enhancements for more cohesion throughout the manuscript. Specifically, the research question and the objectives are not clearly stated making the story they want to highlight a little bit blurry. Additionally, tying those objectives with the methods and results needs more work as I am not sure what activities have been done to address which objectives. The authors may also want to correct the discussion accordingly.

Response Thank you very much for your comments. We modified some parts of the manuscript for better consistency throughout the Objectives, Methods, Results and Discussion sections. We hope this is clearer now.

L1 - Title. Could be shortened to “Land use is more of a determinant than community size of ecological network complexity”.

Response: Thank you very much. The recommended title is shorter and more general to a larger readership. We modified it accordingly.

L4 - Short title. Typo on “networks”; change “on” to “with”.

Response: Thank you very much. We changed these words, as suggested.

Abstract. The implications of your results are missing: we have the results, but what they are supposed to mean? I suggest the authors connect the results directly with their interpretation and/or implications.

Response: We implemented this in the Conclusion paragraph.

L23: variation of what?

Response: Thank you very much. We changed “variation in these networks” to “variation of these networks”

L25-27: Is the whole network comprised of exotic plant species or is there only five exotic species in the whole network system?

Reponse: Response: The network is composed of five species of plants, which are all exotic. We changed the phrase to the following: “. … the effects of land use and number of trophic levels on the interaction networks between five exotic legume plants and the arthropods associated to the seeds of those plants. …”

L25: are seed beetles and Bruchinae the same taxon? If so, make sure to make that clear here as we are not all insect specialists to connect the dots between them.

Response: This was done. We modified the phrase, as shown before.

L26: what are the four land use types?

Response: We added the names of the four land use types

L33-46: I suggest removing or reducing statistical reports to the minimum. I suggest focusing on the biological meaning of those statistical significances and the technical terms (connectance and IE) in the abstract to highlight your results and their implications and make it more accessible for non-network specialist scientists.

Response: We deleted the statistical reports from the main text and added a table with this information. As a result, we think the results are easier to follow.

L40-46: why is the last paragraph separated from the whole abstract?

Response: This was an error. Thank you very much for pointing it out. We deleted it because it did not make sense in the abstract.

L46. is this sentence complete or is there some explanation to it?

Response: It was part of the mistaken paragraph that was deleted.

Keywords. Missing

Response: There is no keyword section in the author guidelines. 

Introduction. Excellent outline and connection of the main concepts of the studies. However, there are too many repetitions of the same information. I suggest that the authors go over the manuscript and make sure one piece of information is stated once and does not repeat it again in another paragraph. As I mentioned earlier, the research question and the objectives of this manuscript are not clear while its importance is well-outlined.

Response: Thank you very much. We revised the entire manuscript and reviewed every section to ensure no repetition of ideas.

L52: a consensus or evidence? Or is there any missing information?

Response: There is consensus between those that have evidence in favor and those that have evidence on the contrary that we are in the Anthropocene. The consensus is that humans are increasingly modifying worldwide ecosystems. The sentence is correct and expresses what we meant to say. 

L54: remove abundance and richness; they are already comprised in functional and taxonomic biodiversity.

Response: Thank you. We deleted “abundance and richness” in the sentence.

L53-55: isn’t the loss of species that creates biodiversity functional loss? You seem to argue the opposite here.

Response: That is not the intended meaning of that sentence. We wrote that those factors have resulted in the loss and extinction of species. The sentence is correct, and we did not change it.

L57-58: remove “in addition to other factors”.

Response: We deleted “in addition to other factors”.

L62: can you advance one or two reasons why the effects of anthropogenic activities on plant-herbivore system have been disregarded despite their importance in the ecosystem

Response: We removed this sentence from this part and added some references to explain these four lines later in the document.

L65-66: this sentence needs a little more explanation.

Response: We added references to support this. However, we do not believe more explanation is required.

L67: is there a gap of knowledge on the ecology of plant-herbivore and parasitoids, or on the effects of anthropogenic activities on this system? If both are lacking, I suggest expanding more on both to highlight the gap and the importance of addressing it.

Response: Thank you very much for your comment, in order to response it, we added some additional information (lines 55-61)

L68-70: As this is an introductory sentence, maybe add another reference that is not focused on plant-herbivore network, such as plant-pollinator or plant-seed disperser network.

Response: We included new references about it (line 64)

L70-72: not clear.

Response: We disagree with the reviewer. We did not change the sentence. Neither of the other two reviewers commented about this.

L74-76: maybe synthetize reference 13, 17, 23-25 into one sentence or two to show the change in ecological networks in response to the introduction of exotic species instead of just citing them.

Response: Again, we decided to leave the sentences as they are. We do not think it is necessary to explain this further.

L77: cite both the original reference 26 and its erratum.

Response: This has been done.

L779-89: why do you come back to the impact of anthropogenic activities? I suggest this paragraph focusing on plant-herbivore network, what is known, and what is lacking, then introducing the lack of investigation in this field and the contradictions you found in the existing studies. Also, make sure to introduce the meaning of specific terms such as connectance. Finally, reorganize the ideas to avoid the back and forth between mutualistic networks and plant-herbivore networks.

Response: We put this paragraph back because we cannot write about the effect of anthropogenic changes in interaction networks without describing what an ecological network is and its advantages for analyzing community structure. Thus, we left the order of the paragraphs as it was previously. However, we did add Lines 71 –73 and 81-84 to describe the relevance of studying antagonistic interactions.

L90-91: any interaction networks? Or just plant-herbivore networks?

Response: All of them 

L99-113: please move the information about the study site, study species in the Materials and Methods, and replace this last section with the goal and objectives of the manuscript t as well as the study system you investigate. Maybe consider also explaining why Legumes are a big deal in this ecosystem. 

Response: Thank you very much. We made the changes as suggested. Lines 124-131 explain why legumes are important to this ecosystem.

L113-116: can you explain briefly how and add relevant references? Also, make sure to link this information about natural enemies in the rationale of your study design.

Response: This is partially explained in the two lines that follow that statement. We included additional lines regarding the natural enemies in the rationale of the study.

L116-120: please reorganize those sentences with the goals, objectives, study system above. Also, make sure to highlight why you specifically chose the arid zone among the other zones you mention in the study site section. Finally, I suggest you add a sentence or two to explain why you chose to examine two types of network system PSP and PSPP.

 “Please reorganize those sentences with the goals, objectives, study system above.” 

Response: This has been done.

“Also, make sure to highlight why you specifically chose the arid zone among the other zones you mention in the study site section” 

Response: This has been done.

“Finally, I suggest you add a sentence or two to explain why you chose to examine two types of network system PSP and PSPP.” 

Response: This is explained in the paragraph in Lines 100 to 103. We also added Lines 94-96 at the beginning of the paragraph outlining the objectives.

Materials and Methods. It is a very concise section although some key points are missing. However, I don’t think that is appropriate for such a technical manuscript. I suggest adding more details on some aspects of the methods, especially the data analysis (detailed below) and restructure the process following the research goal and objectives.

L126-127: not clear and cite only the relevant references.

Response: We changed the sentence and cited just the most relevant reference. 

L128-131: why is this information relevant here? If it denotes the different ecosystem of sampling sites, please say so.

Response: We changed this section entirely to make it clearer.

L134: sampling of what? 

Response: We changed the title to “sampling and identification of organisms”.

L135-136: I am not sure I understand whether you sampled in the vegetation zone or the arid zone, or these two are the same.

Response: We changed the description of the area of study and the methods to make it clearer.

L136: repetitive information.

Response: We changed this section entirely to make it clearer.

L136-138: in how many sites did you sample exactly, five or ten? You say ten but only listed five.

Response: We changed the entire description of sampling to make it clearer.

L138-141: is there any particular reason to choose only these five species?

Response to all four comments: Thank you very much. After reviewing your comments, we realized that this paragraph was not clear enough; therefore, we changed most of it. In addition, as we mentioned in the first version of the manuscript (Line 140), the seed pods were collected according to their availability. Thus, we did not choose these five species. They were collected because those species were the species that had mature seedpods during the time the field work was conducted.

L146-147: 250 g of seeds from which species? 

Response: We changed the phrase to “we obtained approximately 250 g of seeds from each of the species found in each land use type”

L148: please define the acronyms before using them. Also, check other acronyms throughout the text. Are these conditions standardized or is there any reason to choose them?

Response: We revised all abbreviations. “oC” and “RH” are universal abbreviations for degrees Celsius and for relative humidity, respectively, so we did not define these.

L153: delete the “,” between Coleoptera and Bruchinae. I suggest mentioning at the beginning of this section that you sample for those insects specifically as I am not sure if you sample for them specifically or for all arthropods earlier and then you only use identification keys for the Bruchinae here.

Response: We eliminated Bruchinae because the references refer to Coleoptera families and species and Bruchinae, as mentioned in the lines before “The seeds were inspected every 12 hours for approximately 20 days to collect all emerging arthropods”. Then, we wrote: “Preserved arthropods were identified to the lowest taxonomic level”. We believe this is clear enough, so we did not change the paragraph in response to the reviewer’s comment.

L161: Data analysis. Please consider reorganizing the process of analysis. There are too many “first steps, I am not sure which one goes with which one. Also, I suggest to start with the construction of the network, then the process before the actual analyses.

Response: This was done based on the recommendations of the reviewer.

L162-164: Networks of what? Also, how did you characterize the networks from the sampling?: 

Response: We modified the paragraph. In addition we changed “networks” for “interaction networks”

L166-168: What kind of variation? Also, do these land use types correspond to the number of sampling sites you mention earlier?

Response: We changed the paragraph.

L170: Change to “This index compares samples…”

Response: This has been done.

L172-174: It is a bit unclear, please consider revising the sentences. Also, did you use the index for your analysis? If so, please state so. If not, please explain better why this step is critical to your data analysis.

Response: We already wrote “Differences among the metrics of the network in response to land use and community size were calculated using a GLM at p ≤ 0.05. Sampling intensity was used as a covariate because the parameters of the networks could vary in response to the size of the sample”. However, because we changed the paragraph, we think this issue has been resolved. Please see lines 213-225.

L178-180: Change to “This index determines the sensitivity…”

Response: This was done.

L190: Please explain the acronym before using it. Additionally, please make sure to mention the predictors and covariates, how many models did you run, and are there any other steps you took during the comparisons, if any.

Response: This was done. 

L192: You can delete this sentence.

Response: This was done.

Results. Again, good job on the succinct writing composition! Based on the research goal and objectives here, this section fairly addresses them. I suggest revising the section by adding more structure and make the ideas better connected to fully answer the questions. Also, please do not repeat the results in the table in the text, and consider spacing the captions from the main text.

Response: Thank you very much for your comment. We revised the manuscript and made changes and additions for more coherence. 

L196-198: are these species plant or an insect species?

Response: As the phrase suggests, there were five species of plants with 5552 arthropods from six orders, 11 families and 17 species forming 30 interactions. To clarify, we changed the phrase to “five species of plants with 5552 arthropods belonging to six orders, 11 families and 17 species, forming 30 interactions” Table 1 was is added with the species interactions

L198-200: Was the sampling intensity applied to both plant and insect species or plant species only? And between which variables was it compared to (sites, etc.)? Please explain these more in the data analysis section.

Response: It includes both plant and animal species. We provided a more detailed explanation of this index in the Methods section (Lines 213-223).

L200-202: Please write the full value of P if > 0.05. Also, correct throughout the text.

Table 1: Please check to number of sampling sites throughout the text and make sure it is always the same. Sometimes, it is ten, sometimes five, and sometimes eight. Add the caption to the table right after the title and not after the table.

Response: We added the P values from the chi square test. We revised Table 1 accordingly.

L211: Delete the repeated “PSPP”. 

Response: This was done. In fact, we changed the table and revised its description.

Figure 1: Also, please add an annotation of these indications directly on the figure (L218-221).

Response: This was done.

Table 2: Is there any way for this table to be combined with Fig. 1? It is a bit confusing as it is now and you can color-code each insect order for instance.

We examined the figure, and each color code represents a guild. Therefore, we cannot designate a color for each order. We changed Table 1 to make it more informative.

L227-241: can’t the stats be added to Table 1 or another table? It becomes daunting to read. Also, please consider revising the result report. I suggest starting with the PSP community first and report all results related to it, and then the same with the PSPP. 

Response: We included a table with these results. We reported the results to compare the factors we evaluated. Thus, we did not change the sequence of the results.

Maybe also add some basic interpretation of the comparisons for each index here as for now it looks like a repetition of what the stats show.

Response: We changed the tables to make the results clearer and more readable. A basic interpretation of the results has been provided in the Discussion section. 

Discussion. It is a sound and thorough discussion. Similar to other section, please reorganize the ideas to reflect more the research goal and objectives and the results report. Also, it becomes a bit unclear if it is the invasive species or the land use that really threatens Santa Cruz biodiversity in the context of this manuscript. It is ok to have them both here and it is well-thought, but consider restructure the arguments so you avoid repetitions and have better implications on the negative impacts of anthropogenic activities.

Response: Thank you very much. We modified some paragraphs in the Discussion section. Based on these comments, we changed the paragraph on L. leucocephala to make it more consistent with the Discussion and added some lines in the Conclusion too.

L247-248: you did or you didn’t?

Response: The answer is both; in some cases, we did and in others we did not. As the paragraph says, there was no difference in these parameters among the networks when comparing PSP and PSPP. However, we did find differences in the networks with respect to land use. It states the differences between the stages of the two factors evaluated: 1. number of trophic levels, and 2. among land uses.

L248-252: please revise the connection between these two sentences.

Response: We revised the connection between the sentences; it is explained in the following sentences at the end of the paragraph.

L252-254: please make sure to recall your hypotheses/predictions or research questions before saying the results turned out the be as expected.

Response: Thank you very much. We modified the sentence to clarify our meaning (see Lines 278 to 293).

L256-264: to which results this section corresponds to? Please add this before discussing the land use and your proposed hypotheses.

Response: We modified the paragraph because we noticed that it was not in line with the discussion. 

L272: Change “In our case” by “In addition”.

Response: Thank you for the suggestion, but it changes the meaning. We wanted to compare what was reported previously to what we found. Thus, we believe “in our case” suits that purpose.

L272-273: Change to “roadsides had more diverse species and interactions”.

Response: Thank you very much. We modified the text, as suggested, for clarity.

L273-275. This needs references, or mention it if it is a person observation.

Response: Thank you very much. It was modified to clarify that it is a personal observation

L280-282: This is repetitive.

Response: No, it is not repetitive because we are showing that in our results, the connectance value of 1 does not correspond to a well-preserved ecosystem. Instead, it corresponds to the plantation that has only one species of plant. Consequently, we left the sentences as they were previously.

L283: Ok, so what alternative do you propose?

Response: We do not propose any alternative. That is why we wrote the following: “we cannot interpret the significance of this metric to community structure” However, we added some sentences to the paragraph to clarify our ideas regarding this. Please see Lines 320-326.

L292: I suggest you add a paragraph talking about the conservation implications of your findings and/or the importance of addressing the plant-herbivore systems as you pointed out the lack of studies in this field and its importance for the ecosystem.

Response: We expanded the Conclusion paragraph. (Lines 334-342). 

6. PLOS authors have the option to publish the peer review history of their article (what does this mean?). If published, this will include your full peer review and any attached files.

Do you want your identity to be public for this peer review? For information about this choice, including consent withdrawal, please see our Privacy Policy.

Reviewer #1: No

Reviewer #2: No

Reviewer #3: No

General responses regarding the comments made by Reviewer #2:

We are sorry that the reviewer felt frustrated reading the manuscript. The authors also sometimes find science to be frustrating. Especially when we found reviewers that prefer to comment on a rude and aggressive way, and would prefer to comment on how a study should be done in different ways instead of giving comments in a positive way to improve the manuscript. It is challenging to address this type of comments; we would prefer comments that suggest positive ways to improve the manuscript. For example. when writing: “I found these to be frustrating though since there was no attempt to correct these effects for host plant species.” We interpret this comments as: “It is not clear how the metrics used accounted for the effects of host plant species, which is very important since the same species of plants were not found on each land use area”. In fact, we did correct for the effects the reviewer mentioned, but maybe it was unclear which metric we used for that. Our response to this comment and the description of the changes to the manuscript have been provided above.

When writing: “It is not hard to imagine that the trends being ascribed to ‘land-use’ had alot (or at least something) to do with the host-plants present at these sites” we interpreted the comment as: “please provide evidence that the metrics used to describe the networks correct for the differential composition of plant species”. Our response to that comment has already been provided above.

---

## [Decision Letter · Decision Letter 1]

17 Jul 2023

PONE-D-22-17757R1Land use is more of a determinant than number of trophic levels of ecological network complexityPLOS ONE

Dear Dr. Suárez,

Thank you for submitting your manuscript to PLOS ONE. After careful consideration, we feel that it has merit but does not fully meet PLOS ONE’s publication criteria as it currently stands. Therefore, we invite you to submit a revised version of the manuscript that addresses the points raised during the review process.

Two of the original reviewers have once again examined this paper. They agree that it has been substantially improved but still have some minor/slight concerns. These should be addressed in your revision.

We look forward to receiving your revised manuscript.

Kind regards,

Sean Michael Prager, Ph.D.

Academic Editor

PLOS ONE

Journal Requirements:

Reviewers' comments:

Reviewer's Responses to Questions

**Comments to the Author**

1. If the authors have adequately addressed your comments raised in a previous round of review and you feel that this manuscript is now acceptable for publication, you may indicate that here to bypass the “Comments to the Author” section, enter your conflict of interest statement in the “Confidential to Editor” section, and submit your "Accept" recommendation.

Reviewer #1: (No Response)

Reviewer #3: (No Response)

2. Is the manuscript technically sound, and do the data support the conclusions?

Reviewer #1: Yes

Reviewer #3: Partly

3. Has the statistical analysis been performed appropriately and rigorously? 

Reviewer #1: Yes

Reviewer #3: Yes

4. Have the authors made all data underlying the findings in their manuscript fully available?

Reviewer #1: Yes

Reviewer #3: Yes

5. Is the manuscript presented in an intelligible fashion and written in standard English?

Reviewer #1: Yes

Reviewer #3: Yes

6. Review Comments to the Author

Reviewer #1: The authors have made significant changes to the manuscript (now retitled): “Land use is mor of a determinant than number of trophic levels of economic complexity.” In particular, more information is now provided in the main text and in the figures and tables to help communicate the intricacies of the experimental design. Overall, the readability of the manuscript has increased, as has the clarity. One additional comment for the authors to consider in addressing an important aspect of the methodology (sampling period) is below (pertaining to the text in L180-187). Otherwise, I suggest only a few additional minor revisions, as listed below. Thank you.

L64-66. Please review this sentence as there appear to be some errors and as a result, the idea being communicated is not entirely clear.

L171. Does ‘gr’ following 218 refer to grams? If so, the SI unit for grams is ‘g’

L180-187. In the response to the reviewers, the authors indicated that they only sampled for one year due in large part to the need for the permits listed here. The manuscript would benefit from a statement along those same lines in the Methods, as at least two reviewers expressed concerns related to the sampling period. For example, the authors could write (starting on L187) that: “Sampling for this research was restricted to one season (August-September, 2018) due to the logistics involved with procuring permits to sample and export samples from the island of Santa Cruz.”

L199. There is a typo at the start of this sentence, where “This index” is repeated.

L331. Please revise to “… and this may imply that…”

Reviewer #3: By examining an overlooked plant-arthropod network, this manuscript draws attention to the importance of antagonistic interactions in ecosystems as well as the careful steps we should take regarding eradication of invasive species. The authors have substantially improved the content of the manuscript and bring more details to their work. However, the clear rationale of the study design and objectives are still unclear, which tremendously weaken the importance of their result and the implications of such results to the biodiversity of Santa Cruz Island. I recommend the authors to thoroughly restructure the Introduction, reorganize the Methods section, and subset Results and Discussion sections accordingly for a clearer workflow and a more impactful contribution not only to plant-animal and biological invasion ecology. Additionally, I strongly suggest the authors to 1) use shorter sentences to make clearer statements, 2) detail workflow chronologically to avoid repetitions of information, especially in the Methods section, and 3) thoroughly clean and proofread the final version to make sure all components of the manuscript are coherent, complete, non-repetitive, and easier to read. I provide more detailed recommendations below.

Abstract. It has been improved but needs more cohesion between the different sections. Please refer to the guidelines for the information to include for the abstract. Please group information per section as well: background, objectives and hypotheses, methods and analyses, results, and implications, etc. There is currently too many back-and-forth information in this abstract, which leads to confusion. I also suggest removing acronyms as much as possible.

L17-18: Are you looking at the impact of land use of invasive species? I understand both are somewhat related, but you need to highlight what aspects of such relationship or which of them pertains more to your work. Perhaps defining that scheme may help structuring your line of ideas and maintain them throughout the abstract and revise the manuscript accordingly.

L18-22: That is a very long sentence, and it does not help stating the knowledge gap and study objectives. Please reformulate.

L23 & 28: Please use consistent terminology arthropods or Bruchinae throughout the abstract and manuscript as it can be misleading, especially for readers who are not entomologists.

L25: I don’t think you need to detail this here.

L27-30: Please explain the rationale between the use of two different community compositions. Perhaps, removing “three levels” and “more than three levels” and shortening the sentence can help reducing the confusion.

L30-38: Please present concisely the most relevant results related to your objectives here and move the analysis-related information into the appropriate section.

L38-42: This is another long sentence, please split it and separate the implication of your work and the potential impacts of your work.

Introduction. The study rationale still looks blurry to me, which preclude the understanding of the overall importance of this work. The different paragraphs were very much improved but needs a bit more cohesion and details to make sense of the reasoning behind each argument and highlight the actual contribution of the manuscript.

L49: What is the evidence? Please expand a bit your argument here or provide specific examples.

L55-56: What three factors? And how do they modify the structure? Please expand.

L56-57: If you want to contrast the attention received by mutualistic and antagonistic interactions, I suggest start a new paragraph with this sentence and restructure the Introduction accordingly. This is also a great place to highlight how important antagonistic interactions are in different ecosystems.

L61-64: These sentences need a better transition, and please revise it if you consider my suggestion above.

L73-76: Please consider spreading the references to each of the arguments you are using instead of having a long list at the end of the sentence.

L76-77: Isn’t this sentence a consequence of anthropogenic perturbations as well? Why is it separated? Please consider a better transition.

L81: What parameters are you considering here?

L85: “While there was no border effect for such parameters”? Please clarify.

L87-90: This is a very long sentence and has repetitive information. Please reformulate.

L93-99: Is there any reason these details are presented in the Introduction? I suggest removing them or putting the essential information in the Methods section. Also, please remove repetitive information.

L103-106: This is a very long statement of objective. I suggest presenting the objective more concisely and then in another sentence or two present the system you work with to reach this objective.

L106-110: I don’t think this information is necessary in the Introduction. I suggest presenting hypotheses and/or predictions here along with the rationale behind why you want to look at the effects of trophic levels instead. You may also want to add a sentence or two stating the knowledge contribution of this manuscript or a long-term goal of such study.

Methods. There are a lot of repetitive information throughout this section. Additionally, I still can’t make sense of the importance of the activities towards the study objective. As I said in my previous review, please consider stratifying this section. You can do so by 1) briefly introducing the general approach you use to reach your objective, 2) giving separate title to each of the activity so readers can follow which activity pertains to which part of the objective, and 3) using chronological order to describe your activities.

L117-120: I don’t think this piece of information is necessary since you already describe the climate of the “actual” study site further below.

L121: Maybe start a new paragraph with the description of Santa Cruz Island.

L124-127: These sentences need a better transition with the previous and following information.

L127-130: Why coming back to the Galapagos here? Why don't you focus only on Santa Cruz Island?

L130-134: I suggest moving this information in the Introduction when presenting the system you are working with?

L135-141: Please combine this information with the description of Santa Cruz Island earlier and remove repetitive information. Also, consider explaining why you chose Puerto Ayora among many other sites in Santa Cruz Island. The description of the four land use types is also missing from this section, or if there is any, please put them after or with the description of the study site.

L143: This subtitle does not tell much. What “organisms” are you referring to? Please use more informative or descriptive subtitle, perhaps you can put “Seed collection”.

L144-145: Are there any specific species you collect the seeds from? If so, what are these species and why did you focus only on them?

L146-150: I am not sure about the relevance of these sentences here. Please clarify or restructure.

L152-154: Did you choose the trees before the land use types or vice-versa? This needs a clarification as your whole study lays on the impact of land use and if you chose the trees before the land use, it introduces a bias in your study design.

L154-162: This is too confusing, there is too much information to take in all at once. Perhaps a table might help: you can add the land use type with its description and the tree species you sampled there with the number seeds collected. Also, if you want to highlight the importance of Leucaena leucocephala or any other exotic and invasive species, you may want to do so in the description of your study species, which you can put right after the description of the study site.

L168: This paragraph and the next could be under another subtitle. You may put “Collection and identification of Arthropods in the seeds” or something along that line.

L180: Please separate this paragraph from the previous one.

L188: You may also want to chronologically stratify the different steps of your analyses here as there is a lot of back-and-forth in the process.

L190-192: Please consider explaining the differences between interaction network and trophic network. Those terms are sometimes used interchangeably in the manuscript and in other times as different entities.

L193-197: You need to be more specific here. Did you compare the network across the land uses or between the different trophic levels? Or you compared both at the same time? I suggest describing the models you fitted directly here with the independent and dependent variables you used in your analyses.

L198: You may to present why you are looking at these parameters regarding your objective. Then you may want to present these parameters and their description in a table or a more concise way. They are a bit confusing presented like this.

L212: You can include this sentence with the previous two and perhaps in a separate paragraph.

L213-225: How different from L208-211 is this section? I am confused. Perhaps this suggestion can help: you can describe the network first and present any statistical analyses you did along with it, and then compare the network between/across land uses and trophic levels with the statistics.

L221: What does a “GLM at p < 0.05” mean? Please also consider adding the independent and dependent variables of your models.

L224: What is a “null model t model analysis”?

Results. There is much improvement in the presentation of the results. However, the actual link with the objective and the Methods section is still blurry. I suggest presenting the results in accordance with the different subsets of the Methods, which can also help avoiding repetitive information. You may also want to use the same chronological order as in the Methods here to make your results more impactful and clearer. Please do not repeat the figures and tables in the text, but rather provide a little interpretation of the results. Finally, when presenting statistics in the results, please remind us the test you used as you used at least two tests.

L229-231: Please double check the number of arthropods and seeds you collected. There are some contradictions here and in the Methods.

L231-233: Are these legume or arthropod species? A subtitle of the results may help in indicating the reader what species you are talking about here.

L235: Perhaps this could be another subtitle or a new paragraph, depending on the restructuration you bring to the Methods.

L237-241: Perhaps you could combine these two sentences and highlight directly which of the results did not have significant differences instead of separating them, it’s a bit confusing.

Table 1. Associations between plants or seeds? You may want to be clear about that because it can lead to a confusion. You can delete “Species of arthropods … Island of Santa Cruz” as it is redundant to the table title.

Table 2. Please move the title before the table. Also, there are two exact same table 2 in this text, please make sure you only have one. You may also want to merge the cells indicating the same land use types as having two consecutive rows with the same contents are confusing. What p-values are you presenting here? What are you comparing here? Remind us of the statistical test you used in the comparisons.

Table 3. Which covariates and metrics are you examining here? To which models are you referring to? Remind us of the statistical test you used and please define F, P, and DF as they are not informative at all. You may also add a bit explanation of this table in the tex.

Figure 1. Are the parasitoids and phytophagus associated with the arthropods or the plants or the seeds? You may want to explain that when you present your choice to examine the effects of trophic levels. Also, what is LSPPP and LSP? Are they related to PSP and PSPP or different systems? Please explain.

L270-282: I suggest you remain consistent with the structure of your results. Either you describe the results and provide a concise interpretation for all of them, or you only present them.

L277: What does “the networks were functionally redundant” mean? What aspects of your results allow you to state so? Please explain, and I suggest putting such speculation in the Discussion.

L279-280: This sentence does not inform much about your results; either it misses some interpretation, or it needs a better transition with the previous and following statements.

Discussion. As you revise the structure of your Methods and Results section, please also consider this section to follow the different subsets you are using.

L285: Please stay consistent with the wording: arthropods, insects, or Bruchinae.

L288: Did you already explain what you mean by community size in the Introduction or Methods Section? If not, please do so.

L293: “Preserved and continuous forests”

L296-299: This interpretation (or implication?) needs a better formulation. What did you consider, and what does that imply?

L301-302: How relevant is this sentence here? Please revise the transition with the other information.

L307: Preservation of what?

L309: Which hypotheses are you referring to?

L311-313: Why are you comparing seed dispersal interaction to an antagonistic interaction? I don’t think this is relevant here as the two interactions have completely different dynamics and structure, thus your comparisons might not be valid even though both are impacted by land use types.

L317-318: What species and interactions?

L320-321: This closing argument contradicts your whole paragraph. Is this intentional or is there any information missing?

L322: Better briefly describe the system of this study.

L327-328: High value of what?

L330-333: Please consider a better transition with your results here.

L340-343: I am not sure about the relevance of this statement here. Please consider a better transition.

L345: Start of what actions?

7. PLOS authors have the option to publish the peer review history of their article (what does this mean?). If published, this will include your full peer review and any attached files.

Reviewer #1: No

Reviewer #3: No

---

## [Author Response · Author response to Decision Letter 1]

20 Nov 2023

Dear Editor,

Thank you very much for all comments we obtained from the reviewers. Following you will find our response to each comment. The reviewer comment is in bold letter and our response follow the comment in normal lettering.

We look forward to receiving your revised manuscript.

Kind regards,

Sean Michael Prager, Ph.D.

Academic Editor

PLOS ONE

Journal Requirements:

Reviewers' comments:

Reviewer's Responses to Questions

Comments to the Author

1. If the authors have adequately addressed your comments raised in a previous round of review and you feel that this manuscript is now acceptable for publication, you may indicate that here to bypass the “Comments to the Author” section, enter your conflict of interest statement in the “Confidential to Editor” section, and submit your "Accept" recommendation.

Reviewer #1: (No Response)

Reviewer #3: (No Response)

2. Is the manuscript technically sound, and do the data support the conclusions?

Reviewer #1: Yes

Reviewer #3: Partly

Response: Based on the comments made by reviewer # 3, we believe that “Partly” refers to the conclusion section because he/she answers Yes to everything else.

3. Has the statistical analysis been performed appropriately and rigorously?

Reviewer #1: Yes

Reviewer #3: Yes

4. Have the authors made all data underlying the findings in their manuscript fully available?

Reviewer #1: Yes

Reviewer #3: Yes

5. Is the manuscript presented in an intelligible fashion and written in standard English?

Reviewer #1: Yes

Reviewer #3: Yes

6. Review Comments to the Author

Reviewer #1: The authors have made significant changes to the manuscript (now retitled): “Land use is mor of a determinant than number of trophic levels of economic complexity.” In particular, more information is now provided in the main text and in the figures and tables to help communicate the intricacies of the experimental design. Overall, the readability of the manuscript has increased, as has the clarity. One additional comment for the authors to consider in addressing an important aspect of the methodology (sampling period) is below (pertaining to the text in L180-187). Otherwise, I suggest only a few additional minor revisions, as listed below. Thank you.

Thank you very much for your comments. They contributed greatly to improving the quality and clarity of the manuscript. Concerning the comment on the inclusion of the sampling period in the text (L180-187), we chose not to make this change for two reasons. First, we have already addressed the sampling period at the beginning of the Materials and methods section. Second, lines 176-185 specifically refer to the permits obtained for collecting the samples.

L64-66. Please review this sentence as there appear to be some errors and as a result, the idea being communicated is not entirely clear.

Response: Thank you very much. Yes!! You are right. We missed some words in the new version. It is now: “Understanding how the structure of these relationships functions, changes, and is constituted, as well as how it is affected by anthropogenic activities, is necessary for understanding the cascading impact on ecosystem integrity.”

L171. Does ‘gr’ following 218 refer to grams? If so, the SI unit for grams is ‘g’

Response: Thank you. We changed it to “g”.

L180-187. In the response to the reviewers, the authors indicated that they only sampled for one year due in large part to the need for the permits listed here. The manuscript would benefit from a statement along those same lines in the Methods, as at least two reviewers expressed concerns related to the sampling period. For example, the authors could write (starting on L187) that: “Sampling for this research was restricted to one season (August-September, 2018) due to the logistics involved with procuring permits to sample and export samples from the island of Santa Cruz.”

Response: Thank you very much. We added a minor change as suggested. The text now reads: “Sampling for this research was restricted to August-September 2018 due to the logistics involved with procuring permits to sample and export samples from the island of Santa Cruz.”

L199. There is a typo at the start of this sentence, where “This index” is repeated.

Response: Thank you. We deleted the repeated words.

L331. Please revise to “… and this may imply that…”

Response: thank you. It read awkward. We changed to: Thank you. The passage read awkward. We changed it to: “However, one possible way to explain this metric is through a study that demonstrated that networks containing invasive species are more resilient to extinction events [83], suggesting that eradication of the invasive…”

Reviewer #3: By examining an overlooked plant-arthropod network, this manuscript draws attention to the importance of antagonistic interactions in ecosystems as well as the careful steps we should take regarding eradication of invasive species. The authors have substantially improved the content of the manuscript and bring more details to their work. However, the clear rationale of the study design and objectives are still unclear, which tremendously weaken the importance of their result and the implications of such results to the biodiversity of Santa Cruz Island. I recommend the authors to thoroughly restructure the Introduction, reorganize the Methods section, and subset Results and Discussion sections accordingly for a clearer workflow and a more impactful contribution not only to plant-animal and biological invasion ecology. Additionally, I strongly suggest the authors to 1) use shorter sentences to make clearer statements, 2) detail workflow chronologically to avoid repetitions of information, especially in the Methods section

Response: We do not agree with this comment from the reviewer. The methods are presented in a sequential manner, both chronologically and by moving from general to specific activities and procedures. This sequence is organized into three sections: first, we characterize the 'Study site'; second, we describe the 'Sampling and identification of plants and arthropods'; and third, we outline the 'Data analysis'."

Within the 'Sampling and identification of plants and arthropods ' section, we follow a sequence of steps: first, we describe when we performed the sampling; second, we explain how we selected the sampling areas; third, we outline the process of selecting the trees for seed collection within each area; fourth, we detail the seedpod collection process, followed by seed collection and data gathering from each sampled tree; fifth, we describe how we transported the collected seeds to the laboratory, monitored them, and identified the emerging insects. Finally, we conclude this section with information regarding the permits required to conduct the complete study.

Within the 'Data analysis' section, we follow a specific sequence: first, we describe the construction of the interaction networks; second, we define and explain the parameters used in the networks and how we validated the observed values obtained. Finally, we explain how we compared the metrics of the networks.

and 3) thoroughly clean and proofread the final version to make sure all components of the manuscript are coherent, complete, non-repetitive, and easier to read. I provide more detailed recommendations below.

Response: Thank you very much for the comment. We revised the entire manuscript to improve clarity and coherence. 

Abstract. It has been improved but needs more cohesion between the different sections. Please refer to the guidelines for the information to include for the abstract. Please group information per section as well: background, objectives and hypotheses, methods and analyses, results, and implications, etc. There is currently too many back-and-forth information in this abstract, which leads to confusion. I also suggest removing acronyms as much as possible.

Response: We revised the abstract in accordance with the journal's guidelines, which do not require grouping information by sections. As per the guidelines, the abstract should describe the main objectives of the study, provide an overview of the study's methods without going into methodological detail, summarize the key results and their significance, and not exceed 300 words. However, we made sure to incorporate the elements mentioned by the reviewer where they were relevant. We did not include a hypothesis since we did not have one prior to the study, and it would not be accurate to introduce one after the study was completed. Additionally, we adjusted the abstract to adhere to the 300-word limit.

L17-18: Are you looking at the impact of land use of invasive species? I understand both are somewhat related, but you need to highlight what aspects of such relationship or which of them pertains more to your work. Perhaps defining that scheme may help structuring your line of ideas and maintain them throughout the abstract and revise the manuscript accordingly.

Response: Thank you for the comment. It has us working in the better way to show our study. Lines 17-18 are the first two lines of the abstract. They do no refer to the question the reviewer wrote. However, as the 22-24 stated: “we analyzed the effects of land use and number of trophic levels. 

We did revise the abstract, and changed a part of it to make it more coherent with the entire paper. When revising the coherence of the paper we found this: The objective literally says in the abstract: “We analyzed the effects of land use and number of trophic levels on the interaction networks between exotic legume species and their associated arthropods.” 

Then, in the introduction says: “Studies analyzing how the parameters and structure of networks change when considering simultaneously the number of trophic levels and a variety of Thank you for the comment, it helped us improve our manuscript. Lines 17-18 are the first two lines of the abstract. They do not refer to the question the reviewer asked. However, as lines 22-24 stated, “we analyzed the effects of land use and number of trophic levels.”

We revised the Abstract and changed a part of it to make it more coherent with the entire manuscript. When revising the coherence of the text, we note that the objective is clearly stated in the Abstract: “We analyzed the effects of land use and number of trophic levels on the interaction networks of exotic legume species and their associated arthropods.”

Then, the Introduction states: “Studies analyzing how network parameters and structure change when simultaneously considering the number of trophic levels and a variety of land uses are important for testing the multifactorial causes determining the responses of biodiversity to environmental change. Thus, the objective of this study was to 

analyze the effect of land use and number of trophic levels

on

the structure and complexity of the interaction networks

between

arthropods and plants associated with the seeds of five species of exotic leguminous trees…”

This is followed by:

“To do this, we defined two communities that differ in the number of trophic levels on four land uses.”

We changed some phrases, especially the order, of some parts of the Methods section to align it with the introduction, objectives and methods: we defined what four land uses we identified and described them, and in the analysis section, we presented the two communities we compared: three trophic levels vs. more than three trophic levels.

In the Results section, there is a paragraph that already reads: “Hereafter, the results are described for each metric regarding the two factors evaluated: number of trophic levels and land use.” Thus, we described the effects in that regard.

In the Discussion section, we start by writing: “Our study analyzed the effect of land use and number of trophic levels on the structure and complexity of the interaction networks of insects and plants associated with the seeds of five species of exotic leguminous trees…”

Finally, the concluding paragraph states: “In conclusion, the most relevant factor responsible for the differences in the structure of arthropod communities associated with the investigated five legume species was land use type rather than the number of trophic levels involved in the analysis.”

Thus, we are confident that the modifications we made show a clearer progression of ideas, which are maintained from the Abstract to the Conclusion.

L18-22: That is a very long sentence, and it does not help stating the knowledge gap and study objectives. Please reformulate.

Response: We split the sentence into two parts, aligning it with the modifications we made to the abstract.

L23 & 28: Please use consistent terminology arthropods or Bruchinae throughout the abstract and manuscript as it can be misleading, especially for readers who are not entomologists.

Response: Thank you for your comment. We have revised the entire manuscript accordingly.

L25: I don’t think you need to detail this here.

Response: Thank you for the comment. We decided to leave this sentence as it was, because it is part of the methods.

L27-30: Please explain the rationale between the use of two different community compositions. Perhaps, removing “three levels” and “more than three levels” and shortening the sentence can help reducing the confusion.

Response: It is a part of our aims: to determine differences in the networks due to communities of different sizes as it is explained in the sentence: “We analyzed the effects of land use and number of trophic levels on the interaction networks of exotic legume species and their associated arthropods” in the Abstract. We also explain this in the Introduction.

L30-38: Please present concisely the most relevant results related to your objectives here and move the analysis-related information into the appropriate section.

Response: We decided to leave the results as they were. It is clear that we stated the main results in this part of the abstract.

L38-42: This is another long sentence, please split it and separate the implication of your work and the potential impacts of your work.

Response: As we adjusted the abstract to the number of words required, this sentence was also changed.

Introduction. The study rationale still looks blurry to me, which preclude the understanding of the overall importance of this work. The different paragraphs were very much improved but needs a bit more cohesion and details to make sense of the reasoning behind each argument and highlight the actual contribution of the manuscript.

Response: This comment and some others made by the reviewer contradict the evaluation given by the reviewer in question #5: Is the manuscript presented in an intelligible fashion and written in standard English?

The answer was yes. This means that the manuscript was presented in an intelligible fashion. Reviewer #1 answered the same way. Thus, the reviewer answered yes to question #5, but in the comments and regarding the methods the reviewer suggests that the rational is blurry.

L49: What is the evidence? Please expand a bit your argument here or provide specific examples.

Response: This is explained already in lines 50-52 in the original version. Thus, we decided to leave the paragraph as it is.

L55-56: What three factors? And how do they modify the structure? Please expand.

Response: The three factors are mentioned at the beginning of the sentence: Fragmentation, habitat loss and exotic species introductions. This comment has the same problem as any other comments we have already answered: it is based only on the sentence without considering the sequence of ideas in the paragraph. In addition, the Introduction comprises 3.25 pages, which is an average length for this type of study. If we explain and expand each sentence, it would be much longer and confusing. The study is directed toward an expert audience. Thus, we believe that too much explanation and expansion of each sentence would be detrimental to the coherence of the text.

L56-57: If you want to contrast the attention received by mutualistic and antagonistic interactions, I suggest start a new paragraph with this sentence and restructure the Introduction accordingly. This is also a great place to highlight how important antagonistic interactions are in different ecosystems.

Response: Thank you for the first part of your comment regarding the comparison with mutualistic interactions. After considering it, we decided to delete that phrase and lines 61-64 of the original version. It did not aid in the development of the ideas. Regarding the mention to highlight how important antagonistic interactions are, that is what we touch on in the following lines of that paragraph.

L61-64: These sentences need a better transition, and please revise it if you consider my suggestion above.

Response: Thank you for your comment. We considered it and deleted the text accordingly.

L73-76: Please consider spreading the references to each of the arguments you are using instead of having a long list at the end of the sentence.

Response: Done.

L76-77: Isn’t this sentence a consequence of anthropogenic perturbations as well? Why is it separated? Please consider a better transition.

Response: It is separated from the preceding sentence because it explains that the changes could also sometimes, but not always, lead to extinction of populations and species.

That is why we write “Such perturbations can even push…”.

L81: What parameters are you considering here?

Response: The parameters of ecological networks. This is the name given to estimators such as connectance, nestedness, etc. And it is explained in the following lines of the paragraph. 

L85: “While there was no border effect for such parameters”? Please clarify.

Response: Again, this is article is intended for an expert public. The example explains the effect on the network parameters of communities exposed to habitat fragmentation.

L87-90: This is a very long sentence and has repetitive information. Please reformulate.

Response: Thank you. The sentence is no longer repetitive and long since we deleted the text referring to mutualistic interactions. We also deleted the reference to mutualistic interactions in these lines.

L93-99: Is there any reason these details are presented in the Introduction? I suggest removing them or putting the essential information in the Methods section. Also, please remove repetitive information.

Response: The reason is that it shows that exotic species and human activities are a major threat to the specific place we are studying: the Galapagos. The entire Introduction states the general effects of such perturbations on biodiversity, and this paragraph shows the context for the area we are studying. If we remove the paragraph, the Introduction would be too general and would say nothing about the challenges the Galapagos Archipelago faces because of those activities. The beginning of the sentence states: “The Galapagos Islands are an ideal model for analyzing the effects of anthropogenic perturbations on interaction networks.” This is not just because the islands are highly diverse but also because they have been subjected to challenging conservation issues because of the main human activities carried out there. Given these reasons, we decided to leave these sentences in the Introduction.

L103-106: This is a very long statement of objective. I suggest presenting the objective more concisely and then in another sentence or two present the system you work with to reach this objective.

Response: We changed the sentence to make the objective clearer. 

L106-110: I don’t think this information is necessary in the Introduction. I suggest presenting hypotheses and/or predictions here along with the rationale behind why you want to look at the effects of trophic levels instead. You may also want to add a sentence or two stating the knowledge contribution of this manuscript or a long-term goal of such study.

Response: We did not include a hypothesis since we did not have one prior to the study, and it would not be accurate to introduce one after the study was completed

Methods. There are a lot of repetitive information throughout this section. Additionally, I still can’t make sense of the importance of the activities towards the study objective. As I said in my previous review, please consider stratifying this section. You can do so by 1) briefly introducing the general approach you use to reach your objective, 2) giving separate title to each of the activity so readers can follow which activity pertains to which part of the objective, and 3) using chronological order to describe your activities.

Response: We started the Methods section by describing the study site; then, in a different subsection, we describe how and where we collected the samples. In a separate paragraph, we describe how we obtained the arthropods, how we preserved them, and later how we identified them. After the subsection on sampling and identification of organisms, there is a a third subsection titled “Data analysis”. In this subsection, we write separate paragraphs describing the type of networks we built, then another paragraph that explains the metrics we estimated for the networks, and a last paragraph describing how we compared the networks. We strongly consider that the methods are described in a sequential and rational way, as recommended by the reviewer. You cannot analyze data without identifying the species, and identification of species cannot be done without sampling the arthropods. Similarly, in the data analysis, there is no possibility to compare metrics among the networks if there is no estimation of the metrics first. This comment by the reviewer does not make sense after she/he answered that yes, the rationale of the manuscript is presented in an intelligible fashion. The methods were changed in some parts after considering the comments of both reviewers.

L117-120: I don’t think this piece of information is necessary since you already describe the climate of the “actual” study site further below.

Response: We consider that the reviewer is mistaken regarding the descriptions. The first one refers to the mean temperatures of the rainy and dry seasonal periods. The other reference to temperature refers to the mean temperature of the arid zone. We thus left this paragraph as it is.

L121: Maybe start a new paragraph with the description of Santa Cruz Island.

Response: Done.

L124-127: These sentences need a better transition with the previous and following information.

Response: We changed the order of sentences and paragraphs and wrote new sentences in this sections. We believe it is clearer now.

L127-130: Why coming back to the Galapagos here? Why don't you focus only on Santa Cruz Island?

Response: Because we changed the subject and started writing about legumes in the arid zone.

L130-134: I suggest moving this information in the Introduction when presenting the system you are working with?

Response: You are correct. We moved this information to the Introduction.

L135-141: Please combine this information with the description of Santa Cruz Island earlier and remove repetitive information. Also, consider explaining why you chose Puerto Ayora among many other sites in Santa Cruz Island. 

Response: We changed the sentences and rewrote this part, indicating the reasons to conduct the study in Puerto Ayora.

The description of the four land use types is also missing from this section, or if there is any, please put them after or with the description of the study site.

Response: We moved this information from the Sampling section to the Study site section as recommended.

L143: This subtitle does not tell much. What “organisms” are you referring to? Please use more informative or descriptive subtitle, perhaps you can put “Seed collection”.

Response. We changed the subtitle to “Sampling and identification of plants and arthropods”.

L144-145: Are there any specific species you collect the seeds from? If so, what are these species and why did you focus only on them?

Response: This information is presented at the end of the same paragraph containing these lines. 

L146-150: I am not sure about the relevance of these sentences here. Please clarify or restructure.

Response: We do not understand the rationale of the reviewer’s comment. In some notes above, the reviewer asks for a description and an explanation of why we conducted sampling in the arid zone; the explanation here is about that. Then the reviewer comments that this is not relevant. We consider the sentence to be relevant and decided to leave it as is.

L152-154: Did you choose the trees before the land use types or vice-versa? This needs a clarification as your whole study lays on the impact of land use and if you chose the trees before the land use, it introduces a bias in your study design.

Response: The methods are written in the order we performed the sampling. We sampled trees that were found on all the land uses that existed in the sampling area and those that had mature seedpods. Thus, there is no bias because there were no more land uses in the sampling areas with legume trees and with trees with mature seedpods.

L154-162: This is too confusing, there is too much information to take in all at once. Perhaps a table might help: you can add the land use type with its description and the tree species you sampled there with the number seeds collected. Also, if you want to highlight the importance of Leucaena leucocephala or any other exotic and invasive species, you may want to do so in the description of your study species, which you can put right after the description of the study site.

Response: The table described was included in the first version of the manuscript, it is Table 1. We added this information because it was required by reviewer #3 in the first revision of the manuscript

L168: This paragraph and the next could be under another subtitle. You may put “Collection and identification of Arthropods in the seeds” or something along that line.

Response: This paragraph follows the paragraph before on collecting the seeds in the field. We believe that if we put this information in another section we would have to define a section for field collection, another for transport to the laboratory, another for identification of organisms (plants and arthropods), etc., and there would be too many subsections. We decided to leave this part as is: a different paragraph for each step of the “Sampling and identification of plants and arthropods”.

L180: Please separate this paragraph from the previous one.

Response: Done

L188: You may also want to chronologically stratify the different steps of your analyses here as there is a lot of back-and-forth in the process.

Response: Once again, we disagree with the reviewer. We started this section with a paragraph describing the general steps. Then, we dedicated the second paragraph to explaining how we built the networks, followed by another paragraph indicating what parameters we estimated for the networks, and last, a paragraph on how we compared them. We decided to leave this section as is. In addition, reviewer #1 agrees with us and made no comments on this subject.

L190-192: Please consider explaining the differences between interaction network and trophic network. Those terms are sometimes used interchangeably in the manuscript and in other times as different entities.

Response: We revised the manuscript and corrected it as recommended.

L193-197: You need to be more specific here. Did you compare the network across the land uses or between the different trophic levels? Or you compared both at the same time? I suggest describing the models you fitted directly here with the independent and dependent variables you used in your analyses.

Response: We changed some lines in the paragraphs in this section to make it clearer. In addition, a detailed explanation is given in the last paragraph of the section.

L198: You may to present why you are looking at these parameters regarding your objective. Then you may want to present these parameters and their description in a table or a more concise way. They are a bit confusing presented like this.

Response: We decided to leave the description of the metrics used in the analysis as is, in paragraph form. These are just tree metrics. We consider that this would be a very small table and would not add more clarity; we would just be writing the description in a table instead of presenting it in a paragraph. Instead, we added one concise sentence at the beginning of the paragraph and explained why those parameters were estimated.

L212: You can include this sentence with the previous two and perhaps in a separate paragraph.

Response: Done

L213-225: How different from L208-211 is this section? I am confused. Perhaps this suggestion can help: you can describe the network first and present any statistical analyses you did along with it, and then compare the network between/across land uses and trophic levels with the statistics.

Response: These greatly differ. L208-2011 explains how we validated the parameters we calculated for the networks if they were significantly different from a null model, and L213-225 explains how we compared the metrics between land uses and number of trophic levels. We cannot present the statistical analyses we performed. We explained them. This is the methods section.

L221: What does a “GLM at p < 0.05” mean? Please also consider adding the independent and dependent variables of your models.

Response: We changed the test o “at a significance level of p ≤ 0.05”

L224: What is a “null model t model analysis”?

Response: We changed to: “…, we performed a comparison with null models using the t test”

Results. There is much improvement in the presentation of the results. However, the actual link with the objective and the Methods section is still blurry. I suggest presenting the results in accordance with the different subsets of the Methods, which can also help avoiding repetitive information. You may also want to use the same chronological order as in the Methods here to make your results more impactful and clearer. Please do not repeat the figures and tables in the text, but rather provide a little interpretation of the results. Finally, when presenting statistics in the results, please remind us the test you used as you used at least two tests.

Response: We changed some parts of each section of the manuscript to make it clearer. Most of the statistics presented are summarized in Table 2. The only statistics presented in the text are those testing the effect of sampling intensity among land use types at any number of trophic levels; in that case the manuscript states “X2”, which universally means Chi square. Thus, we did not change the text.

L229-231: Please double check the number of arthropods and seeds you collected. There are some contradictions here and in the Methods.

Response: We double checked the numbers in the Methods and Results, and we did not find contradictions. In the Methods section, we refer to the average grams of seeds obtained from each species, from each land use and to the average number of seeds per species. In the Results section, we report the total number of arthropods and the total number of seeds collected.

L231-233: Are these legume or arthropod species? A subtitle of the results may help in indicating the reader what species you are talking about here.

Response: We changed to “the most abundant species were the beetles.”

L235: Perhaps this could be another subtitle or a new paragraph, depending on the restructuration you bring to the Methods.

Response: Following the order we stated in the Methods, we separated this part to compose a new paragraph.

L237-241: Perhaps you could combine these two sentences and highlight directly which of the results did not have significant differences instead of separating them, it’s a bit confusing.

Response: We changed text and combined the sentences as suggested.

Table 1. Associations between plants or seeds? You may want to be clear about that because it can lead to a confusion. You can delete “Species of arthropods … Island of Santa Cruz” as it is redundant to the table title.

Table 1. 

Response: It is an association between arthropods and plants because we are reporting the species of plants to which the seeds belong. We deleted “Species of arthropods … Island of Santa Cruz”

Table 2. Please move the title before the table. Also, there are two exact same table 2 in this text, please make sure you only have one. You may also want to merge the cells indicating the same land use types as having two consecutive rows with the same contents are confusing. What p-values are you presenting here? What are you comparing here? Remind us of the statistical test you used in the comparisons.

Table 2.

Response: We moved the title before Table 2 and eliminated one of the repeated tables. We also eliminated the repeated names of the land uses in the first column. Information regarding the statistical test used and the meaning of the p values was added to the description of the table.

Table 3. Which covariates and metrics are you examining here? To which models are you referring to? Remind us of the statistical test you used and please define F, P, and DF as they are not informative at all. You may also add a bit explanation of this table in the tex.

Response: The reviewer is correct. Table 3 confuses and is not informative. We eliminated this table.

Figure 1. Are the parasitoids and phytophagus associated with the arthropods or the plants or the seeds? You may want to explain that when you present your choice to examine the effects of trophic levels. Also, what is LSPPP and LSP? Are they related to PSP and PSPP or different systems? Please explain.

Response: At this point, we apologize for not complying with many of the comments made. However, we strongly believe it is clear that when we mean the “number of trophic levels involved in the analysis”, we mean that in some analyses, we included only “plant–seed feeders–parasitoids” (three levels); in others, we included more than three levels or “PSPP”. This is explained throughout the entire manuscript. We changed some parts of the description of the figure to make this clearer.

L270-282: I suggest you remain consistent with the structure of your results. Either you describe the results and provide a concise interpretation for all of them, or you only present them.

Response: We mostly report results in the section you mention. To be consistent, we eliminated the phrase “the networks were functionally redundant”.

L277: What does “the networks were functionally redundant” mean? What aspects of your results allow you to state so? Please explain, and I suggest putting such speculation in the Discussion.

Response: We eliminated this sentence after considering that it will not improve the Discussion.

L279-280: This sentence does not inform much about your results; either it misses some interpretation, or it needs a better transition with the previous and following statements.

Response: This sentence goes along with the paragraph because in the preceding lines, we are reporting the results of each network parameter, and in these lines, we report the results for ISA. The lines do not stand alone. They mention that there was a negative ISA value for a land use and a given number of trophic levels, and the next paragraph explains what the values were for the remaining one. Consequently, we decided to leave the sentence as is.

Discussion. As you revise the structure of your Methods and Results section, please also consider this section to follow the different subsets you are using.

L285: Please stay consistent with the wording: arthropods, insects, or Bruchinae.

Response: Revised and adjusted.

L288: Did you already explain what you mean by community size in the Introduction or Methods Section? If not, please do so.

Response: We explained this in the same line to make a connection with our study.

L293: “Preserved and continuous forests”

Response: Changed as suggested.

L296-299: This interpretation (or implication?) needs a better formulation. What did you consider, and what does that imply?

Response: The sentence is clear: when only number of trophic levels were considered, and after, the text states “the results were similar to those found in other studies; that is, the number of trophic levels directly affected the diversity of the interactions. That is = meaning that.

L301-302: How relevant is this sentence here? Please revise the transition with the other information.

Response: We revised this paragraph and changed to “One of the more conspicuous plant species in the network deserves special mention: the invasive L. leucocephala.” 

L307: Preservation of what?

Response: The phrase should be read complete and not in a fragmented way. We wrote “Because it is an invasive species, it could also negatively affect local populations of native legumes [38] and even become a threat to their preservation on the island.” Thus, we refer to the preservation of the subject in the phrase “populations of native legumes”.

L309: Which hypotheses are you referring to?

Response: We changed the word “hypothesis” to “assumptions”.

L311-313: Why are you comparing seed dispersal interaction to an antagonistic interaction? I don’t think this is relevant here as the two interactions have completely different dynamics and structure, thus your comparisons might not be valid even though both are impacted by land use types.

Response. We found the argument valid. We deleted this comparison. 

L317-318: What species and interactions?

Response: The species and interactions we are analyzing in the study.

L320-321: This closing argument contradicts your whole paragraph. Is this intentional or is there any information missing?

Response: We deleted these sentences.

L322: Better briefly describe the system of this study.

Response: We deleted the sentence because the comparison was out of place.

L327-328: High value of what?

Response: We changed this term to “high connectance value”.

L330-333: Please consider a better transition with your results here.

Response: We changed the transition connector in the phrase.

L340-343: I am not sure about the relevance of this statement here. Please consider a better transition.

Response: We changed the phrase to make it clearer.

L345: Start of what actions?

Response: We eliminated these words. To explain them would be redundant as the paragraph stars by stating “The development of plans for the eradication or control”. Thus, those are the actions we refer to.

7. PLOS authors have the option to publish the peer review history of their article (what does this mean?). If published, this will include your full peer review and any attached files.

Do you want your identity to be public for this peer review? For information about this choice, including consent withdrawal, please see our Privacy Policy.

Reviewer #1: No

Reviewer #3: No

---

## [Editor Report · Decision Letter 2]

22 Nov 2023

Land use is a stronger determinant of ecological network complexity than the number of trophic levels

PONE-D-22-17757R2

Dear Dr. Suárez,

We’re pleased to inform you that your manuscript has been judged scientifically suitable for publication and will be formally accepted for publication once it meets all outstanding technical requirements.

Kind regards,

Sean Michael Prager, Ph.D.

Academic Editor

PLOS ONE
---

## [Editor Report · Acceptance letter]

1 Feb 2024

PONE-D-22-17757R2 

PLOS ONE

Dear Dr. Amarillo-Suárez, 

I'm pleased to inform you that your manuscript has been deemed suitable for publication in PLOS ONE. Congratulations! Your manuscript is now being handed over to our production team.

Kind regards, 

on behalf of

Dr. Sean Michael Prager 

Academic Editor

PLOS ONE